# Chemoselective dual functionalization of proteins via 1,6-addition of thiols to trifunctional *N*-alkylpyridinium

Lujuan Xu[1,2,5], Maria J. S. A. Silva [2,5], Jaime A. S. Coelho [3], Joscha Borho [4], Nicole Stadler [4], Holger Barth[4], Seah Ling Kuan [2] ✉ & Tanja Weil [2] ✉

Chemoselective dual functionalization of proteins has emerged as an invaluable tool to introduce two distinct payloads to proteins, thus greatly expanding their structural and functional repertoire for more advanced biomedical applications. Here, we introduce *N*-alkylpyridinium reagents as soft electrophiles for chemoselective dual modification of cysteine residues in peptides or proteins via a 1,6-addition reaction. The *N*-alkylpyridinium derivatives can be synthesized in two reaction steps revealing good water solubility, high labelling efficiency and chemoselectivity towards cysteine over lysine/*N*-terminal amine residues, even when used in large excess. This reaction can be combined with strain-promoted azide-alkyne click (SPAAC) and inverse-electron-demand Diels−Alder (iEDDA) reactions to achieve dual functionalization of proteins in a sequential simple one-pot reaction. As a proof-of-concept, the Rho-inhibiting enzyme *Clostridium botulinum* C3 is functionalized with a cancer cell-targeting peptide and a fluorescent dye for the inhibition of specific Rho-mediated intracellular pathways. The high stability, ease of synthesis, fast reaction kinetics, high water-solubility and chemoselectivity make *N*-alkylpyridinium reagents unique for dual modification of peptides and proteins to increase their functional diversities for medical applications.

Biologics are emerging as therapeutics for targeted or precision medicine, offering a new avenue for the treatment of serious diseases, including tumor management. The development of biotherapeutics with specific targeting capabilities often requires the preparation of complex biomolecules, combining various functionalities to enable combination therapy, therapy and diagnostics, and/or real-time monitoring[1,2]. For example, there is a growing interest in integrating drugs and imaging agents into therapeutic proteins for disease treatment and simultaneous monitoring of treatment efficacy. In addition, antibody drug conjugates (ADCs) with two or more different drug molecules attached allow combination therapy to overcome drug

resistance and address tumor heterogeneity[2–4]. However, the preparation of these multifunctional protein conjugates has predominantly relied on statistical modifications on the protein surface[5–8], leading to heterogeneous product mixtures with different pharmacokinetic profiles, batch-to-batch variability and compromised protein activity, thus posing substantial obstacles for further in vivo applications[9]. In this context, site-selective dual modification of proteins has emerged, allowing the simultaneous introduction of two different functionalities at a specific site, resulting in homogeneous multifunctional protein conjugates with tailored properties[10]. Consequently, a protein can be repurposed in a precise manner to acquire a

[1]Hangzhou Institute of Medicine, Chinese Academy of Sciences, Hangzhou, Zhejiang 310018, China. [2]Max Planck Institute for Polymer Research, Ackermannweg 10, 55128 Mainz, Germany. [3]Centro de Química Estrutural, Institute of Molecular Sciences, Faculty of Sciences, University of Lisbon, 1749-016 Lisbon, Portugal. [4]Institute of Experimental and Clinical Pharmacology, Toxicology and Pharmacology of Natural Products, Ulm University Medical Center, Albert-Einstein-Allee 11, 89081 Ulm, Germany. [5]These authors contributed equally: Lujuan Xu, Maria J. S. A. Silva. ✉e-mail: kuan@mpip-mainz.mpg.de; weil@mpip-mainz.mpg.de

combination of two new features, such as a new bioactivity profile, cellular internalization or fluorescence for bioimaging, or disease treatment, among others[11–14].

Nevertheless, due to the variety of reactive groups on the protein surface, it is rather challenging to select two different amino acid residues for dual modification that provide orthogonal reactivities and sufficient surface abundance without using non-canonical amino acids[15–19]. Thus, one plausible approach relies on the design of a trifunctional bioconjugation reagent that already carries two payloads (such as two drugs or fluorescent dyes) or two bioorthogonal groups to modify a specific residue on a protein surface in a late-stage chemoselective fashion[20,21]. Cysteine conjugation is an essential tool in protein research as cysteines occur in low abundance on the protein surface and exhibit high nucleophilicity, which is important for high site-selectivity and reactivity. However, until now, trifunctional bioconjugation reagents targeting cysteine residues[20,22–26] are still much less explored compared to the vast library of reagents for single cysteine modification for several reasons: Firstly, trifunctional reagents often require labor-intensive, multi-step synthesis involving tedious protection/deprotection processes[1,27]. Some bioconjugation reagents offer poor water solubility or steric hindrance[28,29], leading to low modification efficiency, while others may exhibit cross-reactivity to other amino acid residues as these reagents are often used in large excess in protein modification, leading to heterogeneous mixture[30]. For example, maleimide reagents are most commonly used for dual modification at single site, but they can cross-react with the N-terminus or lysine residues when used in a large excess[19,31,32]. Maleimides are also known to undergo side reactions with azides, rendering them incompatible for incorporation in a single, trifunctional reagent[33,34]. Furthermore, one-pot functionalization with minimized purification steps will be essential to reduce product loss and processing time, which is needed for on-site synthesis at a medical site. Thus, structurally simple, stable and easy-to-synthesize trifunctional bioconjugation reagents targeting cysteine residues with excellent chemoselectivity and high labeling efficiency are urgently needed.

Assisted by Density Functional Theory (DFT) calculations, we hypothesize that highly water-soluble and simple-to-synthesize N-alkylpyridinium reagents can undergo 1,6-addition with thiols to achieve chemoselective dual modification of proteins at cysteine residues. Herein, we demonstrated that N-alkylpyridinium derivatives containing alkyne, azide and tetrazine groups can be obtained by a simple and straightforward synthesis with commercially available starting materials. The N-alkylpyridinium reagents can react with thiol groups with excellent regioselectivity, chemoselectivity and fast kinetics. Notably, the reagent is stable to hydrolysis and site-selective cysteine modification of proteins can be achieved, even when used in large excess. This is in contrast to conventional maleimide reagents, which hydrolyze to form unreactive maleamic amides[35] or undergo side reactions with amine groups[31,32,36,37]. Furthermore, the unprecedented 1,6-addition of thiols to the N-alkylpyridinium derivatives for bioconjugation, is compatible with strain-promoted azide-alkyne click (SPAAC) and inverse-electron-demand Diels–Alder (iEDDA) reactions, thus allowing the in situ dual functionalization of native proteins in one-pot. As a proof of concept, *Clostridium botulinum* C3, a model enzyme that catalyzes the ADP-ribosylation of Rho proteins[38,39] and is only naturally taken up by macrophages and monocytes[40,41], was dually modified with a fluorescent dye (Cy5) and a cell-targeting peptide via two bioorthogonal reactions. In this way, the C3 enzyme was able to reach Rho proteins as intracellular drug targets to alter the morphology of cancer cells, expanding the evolutionary optimized features of C3 toxins for potential biomedical applications.

## Results and Discussion
### Chemical design of N-alkylpyridinium reagents
We used the commercially available (E)-3-(pyridin-4-yl)acrylic acid bearing a conjugated double bond, a carboxylic acid and a pyridine, to design a class of trifunctional reagents for site-selective dual modification. Quaternization of the nitrogen atom of the pyridine increases the electrophilicity of the conjugated double bond in the *para* position through electron density delocalization. Therefore, based on the hard-soft acid-base (HSAB) theory by Pearson, we hypothesize that the N-alkylpyridinium reagents (Fig. 1) that have a more extended conjugated system and diffused electron density are soft electrophiles, which prefer to react with soft nucleophiles, such as thiolates rather than amines (hard nucleophiles)[42]. Thus, we anticipate that the 1,6-addition of thiols to N-alkylpyridinium will provide higher chemoselectivity compared to the conventional 1,4-thiol addition[43–45].

To design the reagent, the impact of N-quaternization and different substituents at the carbonyl group on the selectivity of the addition reaction at the double bond was first estimated based on the Fukui indices (Fig. 2a and Supplementary Fig. 18). The Fukui index is calculated by DFT to determine the electrophilicity or nucleophilicity of the different atoms in a molecule. Higher Fukui index f+ indicates increased electrophilicity of a specific atom[46]. For the carbon atoms at the double bond of (E)-3-(pyridin-4-yl)acrylic acid, the calculated Fukui indices of 0.16:0.17 ($C_{blue}:C_{pink}$, Fig. 2a) indicate that the reagent would undergo non-regioselective addition reaction. Quaternization of the nitrogen (0.04:0.17), as well as amidation of the carboxylic acid (0.01:0.17) favors the 1,6-addition reaction, whereas other substituents such as ester, ketone or nitrile groups have no positive effect on the Fukui indices (Supplementary Fig. 18). Moreover, due to the positive charge of the N-alkylpyridinium group, the corresponding bioconjugation reagents reveal lower partition coefficients (Octan-1-ol to water, cLog $P_{o/w}$), in comparison to other known reagents (Supplementary Fig. 1). This indicates higher water solubility, which could also reduce or prevent protein aggregation and denaturation of the bioconjugates[47,48].

Accordingly, we employed amide functionalization of the carboxylic acid group, followed by N-methylation on the commercially available (E)-3-(pyridin-4-yl)acrylic acid to prepare compound **1** (Fig. 2c). To explore the versatility of N-alkylpyridinium reagents for site-selective dual modification of proteins, different reactive handles such as azido, alkyne, or tetrazine groups were also incorporated to offer N-alkylpyridinium derivatives **2**–**6** in moderate yields (up to 60%). Known protein dual modification reagents often require multistep synthesis; i.e. the allyl sulfone reagents require five-step synthesis[25]. Maleimide-based trifunctional reagents containing two additional bioorthogonal groups for dual modification, require multiple deprotection steps of Boc groups[27]. In contrast, the N-alkylpyridinium derivatives can be readily obtained in high yields in only two reaction steps, without complicated protection and deprotection procedures or tedious purification steps, which facilitates practical applications for a broader community.

### Reaction profile of N-alkylpyridinium with nucleophiles
The chemoselectivity of N-alkylpyridinium towards different nucleophiles such as thiol or amino groups was first investigated with compound **1** due to its structural simplicity for analysis and characterization. First, activation free energies ($\Delta G^{\ddagger}$) for the addition reactions between compound **1** and model thiolates (MeS⁻) and amines ($MeNH_2$) are predicted by DFT calculations (Fig. 3a and b). The data indicates that the activation free energies for the reaction between compound **1** and thiolate ($\Delta G^{\ddagger}$ = 7.6 kcal/mol) is much lower (by 10.1 kcal/mol) than that for amines ($\Delta G^{\ddagger}$ = 17.7 kcal/mol) (Fig. 3a and b). Therefore, we anticipate an improved chemoselectivity towards thiols over amines, including the N-terminus, based on the significant difference in the calculated activation barrier. Next, we validated the DFT calculation by using compound **1** (1 equiv) to react with thiol-containing **7** (2.1 equiv) and amine-containing compound **8** (2.1 equiv) in acetonitrile (ACN): phosphate buffer (PB) (pH 7) (v/v = 2:3). The reaction progress was monitored using high performance liquid chromatography (HPLC) and liquid chromatography-mass spectrometry (LC-MS). Only the thioether

derivative **9** was formed with quantitative conversion (Fig. 3d). Besides the observed reaction with thiol groups, compound **1** did not show any cross-reactivity with other amino acid residues that contain nucleophilic side chains, such as tyrosine, methionine, arginine and others under the same reaction conditions (Fig. 3e, Supplementary Fig. 4). Next, we investigated the influence of pH on the thiol addition reaction. Quantitative conversion to product **9** was observed after 4 h at pH 6–9 (Fig. 3f), showing the efficiency and the robustness of the reaction. There was a decrease in conversion to around 75% at pH 5, probably due to the lower amount of thiolate anions under more acidic conditions (Fig. 3f, Supplementary Fig. 6). Next, the reaction conversion was investigated over time with the same model reaction at pH 7. Compounds **1** and **7** (2 equiv) were incubated in ACN:PB (pH 7) at room temperature and the reaction was monitored by HPLC at different time intervals. The data showed that almost quantitative conversion was already achieved within 2 h of the reaction (Fig. 3g, Supplementary Fig. 8). We also determined that compounds **1**, **5** and **6** are soluble in aqueous media at high concentrations (10–50 mM) and the reactions can also be conducted without organic solvent if required.

Besides reactivity, sufficient stability of *N*-alkylpyridinium reagents in solution is also an important feature for the subsequent bioconjugation reaction. Therefore, the stability of compound **1** was evaluated at three different pH values (pH 6, 7 and 8) for 24 h. According to the HPLC data, no decomposition of compound **1** was observed for up to 24 h (Supplementary Fig. 9–11). In contrast, maleimide reagents can easily hydrolyze to unreactive maleamic amides, especially at basic pH ($t_{1/2} < 2$ h). Thus, the *N*-alkylpyridinium reagents are superior to maleimides if exact quantities have to be used for bioconjugation under stoichiometric control or if extended reaction time is needed when targeting less accessible thiols i.e. on proteins.

## Regioselectivity of the *N*-alkylpyridinium derivatives with NMR studies and DFT calculations

The thiol-addition of **7** to **1** can potentially afford two products, **9** and **9'** (Fig. 4a), which are indistinguishable by MS analysis. Thus, the regioselectivity was investigated with nuclear magnetic resonance (NMR) spectroscopy using compound **9**. A comparison of the proton NMR of **1** and **9** indicated that the peaks at 8.64 ppm and 8.13 ppm assigned to the vinylic protons of **1** are absent in the product, demonstrating that the reaction took place at the double bond. The details and assignment of the proton NMR spectrum of the isolated product **9** are given in Supplementary Fig. 13, which confirms the formation of a single product, rather than a mixture of different regioisomers. The Heteronuclear Single Quantum Coherence (HSQC), Heteronuclear Multiple Bond Correlation (HMBC) and $^1$H-$^1$H correlated spectroscopy (COSY) spectra of compound **9** were acquired to further confirm the site of the thiol-addition reaction ($C_5$ or $C_6$) (Fig. 4a, Supplementary Fig. 14–17). HMBC data showed two cross peaks between $C_3$ and $H_5$ (Fig. 4b), indicating the thiol group was added to the vinylic $C_6$ in a 1,6-addition reaction. No single cross peak was observed between $C_3$ and $H_5$ in the HMBC spectrum, confirming that the thiol-addition did not occur at $C_5$. Furthermore, both $C_4$ and $C_7$ showed a single cross peak with $H_6$ but two cross peaks with $H_5$ in the HMBC spectrum, which further confirmed the 1,6-regioselectivity (Supplementary Fig. 16). The pronounced reactivity preference for the 1,6-addition is in agreement with the relative electrophilicity of each carbon ($C_5$ and $C_6$) in compound **1** based on the calculated Fukui indices (Fig. 2a).

Further DFT calculations were employed to determine the Gibbs free energies of activation for the thiol-addition at $C_5$ and $C_6$ (14.1 kcal/mol and 7.6 kcal/mol, respectively), using methanethiolate as a model

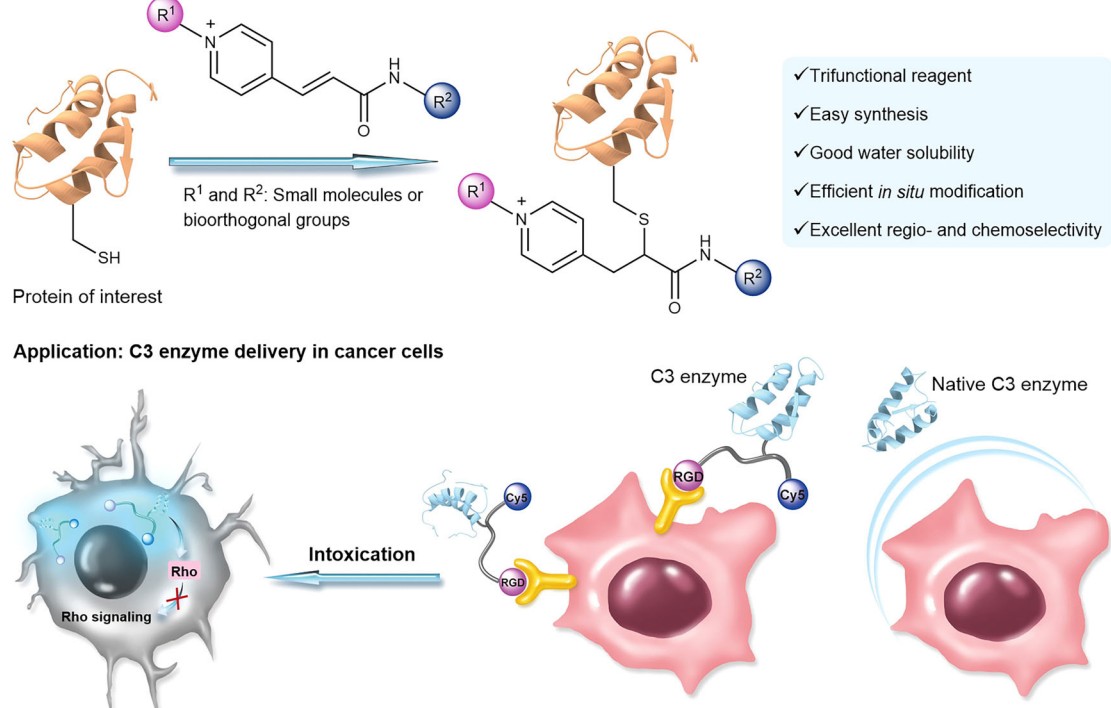

**Fig. 1 | Overview of *N*-alkylpyridinium reagents to achieve dual modification of functional proteins and proof-of-concept application with a toxin enzyme.** The tripeptide arginine-glycine-aspartic acid (RGD) has been selected as the targeting moiety that binds to the integrin receptors on the cancer cell's surface. C3bot1 toxin (C3) from *Clostridium botulinum* that catalyzes ADP-ribosylation of Rho proteins was selected for dual modification with a fluorescent dye (Cy5) and a RGD peptide for in vitro imaging and concomitant inhibition of specific Rho-mediated cellular pathways.

**Fig. 2 | Rational design of *N*-alkylpyridinium reagents. a** Fukui indices for (*E*)-3-(pyridin-4-yl)acrylic acid, after quaternization of the nitrogen atom and amide formation reveals increasing selectivity for 1,6-addition (pink) over the more common 1,4-addition (blue). **b** Model reaction showing the 1,6- and 1,4-addition of thiols to compound **1**, respectively. **c** Synthetic route of the various *N*-alkylpyridinium derivatives (**1–6**, yields up to 60% overall steps). *N*-(3-Dimethylaminopropyl)-*N'*-ethylcarbodiimide (EDC), 4-Dimethylaminopyridine (DMAP).

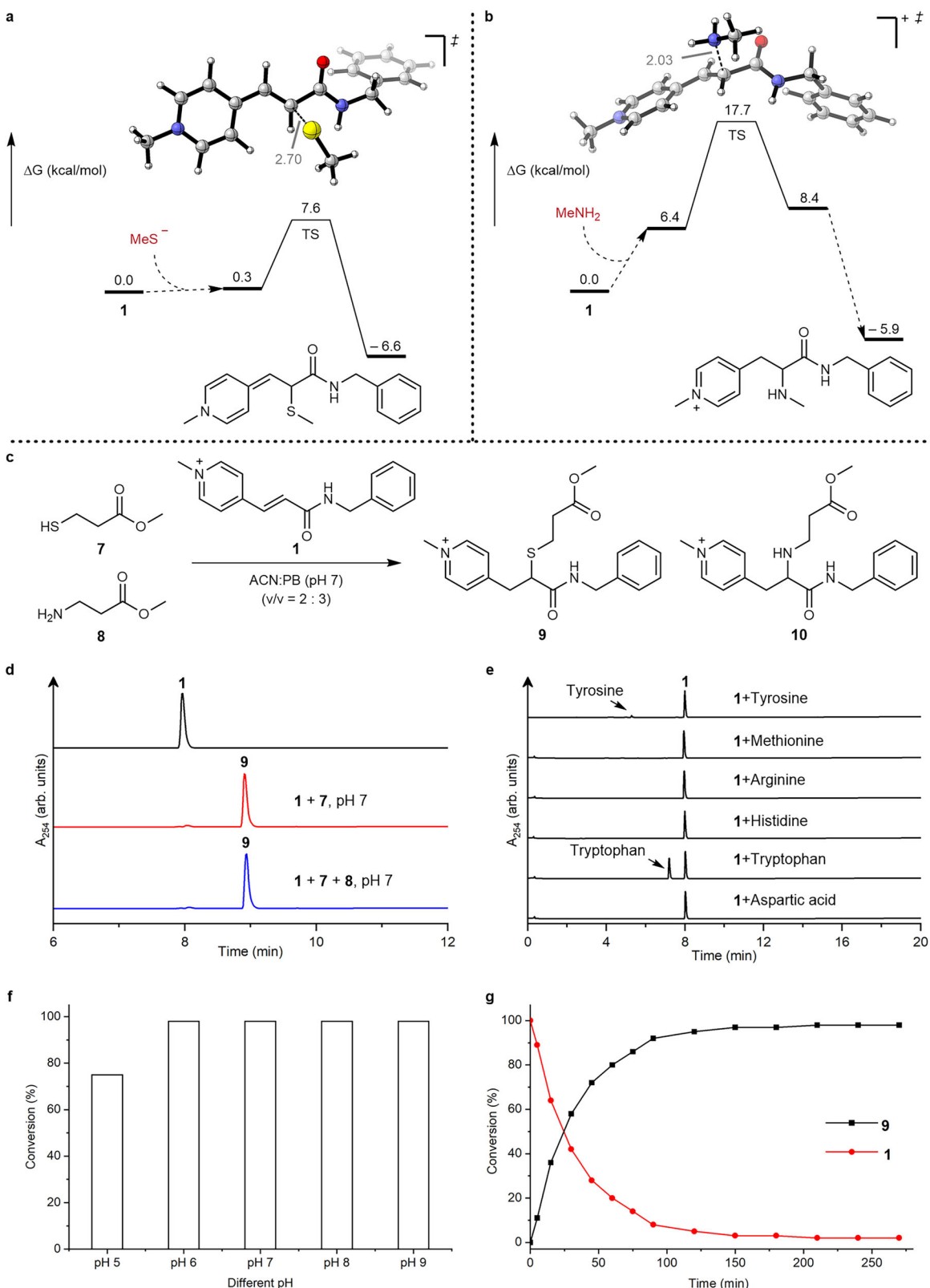

**Fig. 3 | Chemoselectivity studies for *N*-alkylpyridinium reagents. a** Gibbs free energy profile for the addition reaction of methanethiolate anion to compound **1**. **b** Gibbs free energy profile for the addition reaction of amines (MeNH₂) to compound **1**. **c** Model reactions of compound **1** with **7** and **8**. **d** HPLC traces of model reaction between compound **1** and **7** (and **8**) at 4 mM, phosphate buffer (PB) pH 7, after 4 h. **e** HPLC traces of the reactions between compound **1** and different nucleophilic amino acids (tyrosine, methionine, arginine, histidine, tryptophan, aspartic acid). **f** Evaluation of the conversions of reactions between compound **1** and **7** at different pH after 4 h. **g** Reaction profile of compound **1** and formation of compound **9** determined as percentage with reference to the internal standard (Fmoc-Phe-OH) by integration of the HPLC peak. Single HPLC injections were performed for semi-quantitative analysis. **f-g** Source data are provided as a Source Data file.

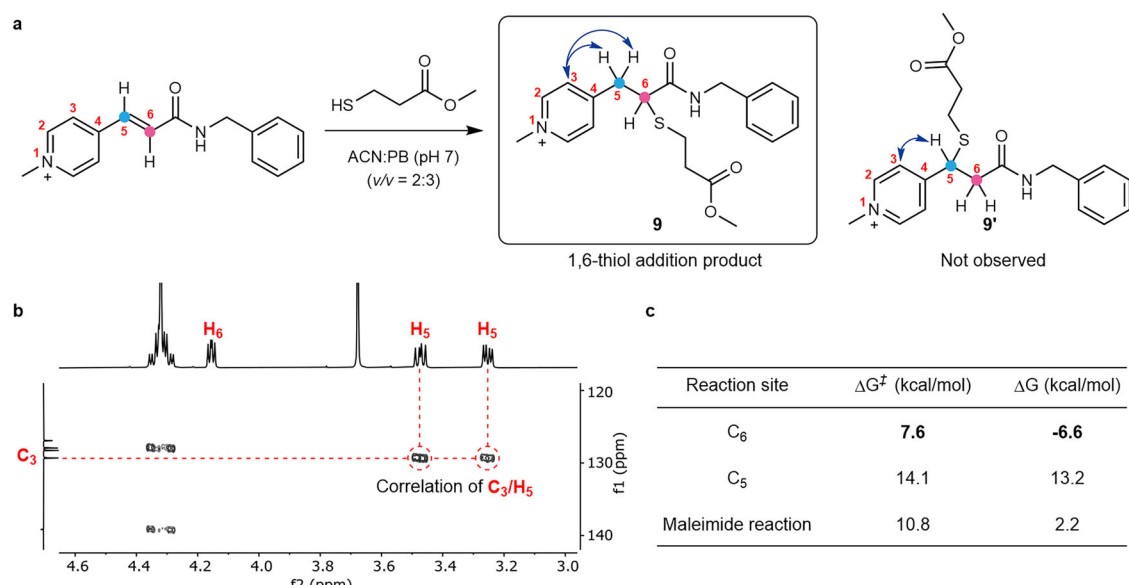

**Fig. 4 | Structure elucidation of 1,6-addition products with *N*-alkylpyridinium reagents. a** Model reaction between **1** and **7** to form two different regioisomers: **9** and **9'** in acetonitrile (ACN):phosphate buffer (PB) (2:3) solution. **b** HMBC spectrum of the obtained thiol-addition product **9**. **c** Gibbs free energies of activation for 1,6-addition of thiols using methanethiolate at $C_5$ and $C_6$ of compound **1** and for 1,4-addition reaction with maleimide (most favored is in bold).

reagent (Fig. 4c, Supplementary Fig. 20). Moreover, studies using neutral methanethiol indicate that both pathways are equally exergonic (Supplementary Fig. 22A). The DFT calculation further corroborated the experimental results that the thiol-addition at $C_6$ is selectively favored and only product **9** was obtained. The calculated activation energy for maleimide conjugation with the same model thiol was determined to be 10.8 kcal/mol, which is comparable to the modification strategy reported here. Calculation of proton affinity (PA) of the 1,6-addition adduct intermediate indicates a preference for protonation at the carbon atom of the former double bond in the final reaction step (Supplementary Fig. 21). Furthermore, the analysis of the computed frontier molecular orbitals of compound **1** reveals that the lowest unoccupied molecular orbital (LUMO), the acceptor orbital, is mainly located at the conjugated pyridinium, while the more energetic LUMO + 2 is located at the conjugated amide (Supplementary Fig. 19). This observation indicates that the carbon atom leading to the 1,6-addition product is more electrophilic, which is consistent with the calculated Fukui indices (Fig. 2a). Taken together, the NMR and DFT studies show that the 1,6-addition is highly regioselective and only yields a single product. This characteristic is crucial for ensuring sample homogeneity and reproducibility, which are of paramount importance for medical applications.

## Site-selective modification of peptides

The model reaction between **1** and **7** used for characterization shows high chemoselectivity and regioselectivity. Next, we used the *N*-alkylpyridinium **1** to modify cysteine residues in more complex structures such as peptides and proteins. PC8, a nuclear targeting peptide containing an accessible cysteine, an *N*-terminus and four lysines in its sequence, was selected to demonstrate the chemoselectivity and regioselectivity of *N*-alkylpyridinium reagents towards thiol *versus* amino groups (Fig. 5a, see SI for experimental). HPLC analysis of the crude reaction mixture of reagent **1** (1 equiv) with PC8 (0.9 equiv) in ACN:PB (50 mM, pH 7, *v/v* = 2:3) shows high conversion to the corresponding 1,6-conjugate **11** after 4 h along with the remaining excess of starting material **1** (Fig. 5b, see SI for experimental). As a control, thiol-reactive reagent 4,4'-dithiodipyridine (4-DPS) was used to mask the cysteine residue in PC8 peptide via a thiol-disulfide exchange reaction.

No reaction occurred between reagent **1** and the 4-DPS modified PC8 peptide, even in the presence of an *N*-terminus and four lysine residues (SI section 8.3). These data clearly show the excellent chemoselectivity and high efficiency of *N*-alkylpyridinium reagents for reacting with cysteine residues in peptides. Other peptides such as the integrin receptor binding peptide RGDC, CEIE peptide, chemokine receptor 4 antagonist WSC02 (sequence in Fig. 5), antimicrobial peptide Tet (sequence in Fig. 5) and pan-coronavirus (CoV) fusion inhibitor EK1C (sequence in Fig. 5) with different lengths, sequences and cysteine positions were also successfully modified with **1** under mild reaction conditions and isolated in moderate to high yields (57–95%, Fig. 5d, SI section 8.1). We further determined the conversions for the peptides (76%–quantitative), which show only a single product formed and no side products. The conversions observed (PC8: 76%; WSC02: 84%) are in a good agreement with the isolated yields (PC8: 65%; WSC02: 86%). Our results corroborate the general applicability of *N*-alkylpyridinium reagents for site-selective modification of peptides at the cysteine residue.

Next, we performed extended stability studies, using HPLC analysis (single injection per sample for semi-quantitative analysis), to determine the advantages and limitations of the conjugate formed with the 1,6-addition to thiol. The PC8 bioconjugate **11** was selected for further stability study, in comparison with PC8 maleimide conjugate (SI-Section S8.2). The stability, for up to 14 days, was carried out under various conditions, including different pHs buffers (6–8) and in the presence of glutathione (1 mM GSH), which is found in the cytosol of cells[49]. From the HPLC analysis, we determined that the stabilities of conjugate **11** and the PC8 maleimide conjugate in buffered solutions are comparable. PC8 maleimide was more stable at pH 6, but is hydrolyzed at pH 7 significantly (~ 20% in 3 days, ~70% in 14 days) and at pH 8 (~20% in 12 h, 100% in 14 days), to form the stable ring opening product. The observation is consistent with the reported literature, where ~70% of the thiosuccinimide conjugates are hydrolyzed via ring opening, and present thiol exchange (Supplementary Fig. 56 and 62) or retro-Michael to a lower extent[50–52]. On the other hand, the PC8 bioconjugate **11** is more stable in acidic and neutral pH, compared to basic conditions (Supplementary Fig. 52–55). The PC8 bioconjugate **11** is most stable at pH 7, and after

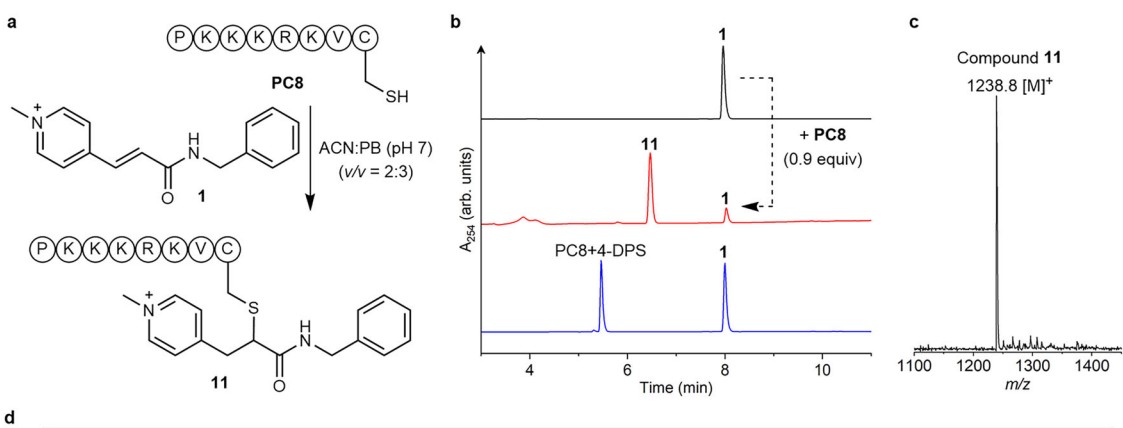

**Fig. 5 | Scope of peptide substrates. a** PC8 peptide modification with compound **1**. **b** HPLC trace of **1**, the reaction mixture of PC8 and **1** (1.1 equiv), and the reaction mixture of DPS-modified PC8 peptide and compound **1** (from top to bottom). Experimental for (**a**, **b**) can be found in SI (Supplementary Fig. 64–66). Single HPLC injections were performed for qualitative analysis. **c** ESI-MS of **11** (calculated: 1238.9 [M]$^+$, found: 1238.8 [M]$^+$). **d** Site-selective modification of different peptides with compound **1**. Reactions were performed in aqueous media (See Methods). Phosphate buffer (PB) and 4,4'-dithiodipyridine (4-DPS). Cysteine residue is in bold.

| Peptide | Sequence | Isolated yield (%) | Calc. (*m/z*) | Obs. (*m/z*) | |
|---|---|---|---|---|---|
| CEIE | NH$_2$-**C**EIE-CO$_2$H | 95 | 373.2 | 373.3 | [M+H]$^{2+}$ |
| RGDC | NH$_2$-RGD**C**-CO$_2$H | 65 | 620.7 | 620.2 | [M]$^+$ |
| PC8 | NH$_2$-P<u>KKKRKV</u>**C**-CO$_2$H | 57 | 702.3 | 702.4 | [M]$^+$ |
| WSC02 | NH$_2$-IVR<u>WSKK</u>VP**C**V<u>S</u>-CO$_2$H | 86 | 827.5 | 827.9 | [M+H]$^{2+}$ |
| Tet | NH$_2$-<u>H</u>LNIL<u>STL</u>WKYR**C**-CO$_2$H | 94 | 1036.0 | 1036.2 | [M+H]$^{2+}$ |
| EK1C | NH$_2$-<u>S</u>LDQINVTFLDLE<u>YE</u>M<u>KK</u>LEEAI<u>KK</u>LEE<u>SY</u>IDLKEL**C**-CO$_2$H | 92 | 1563.1 | 1563.6 | [M+2H]$^{3+}$ |

24 h (72 h) at room temperature, where only <10% (<20%) degradation was observed. We observed that the PC8 bioconjugate (**11**) undergoes more significant 1,6-elimination at neutral to basic than that in acidic pH (~53 to 90% in 14 days at pH 7 and 8, respectively, *versus* ~35% at pH 6).

GSH is present in low micromolar concentration in plasma and are present in higher concentrations in cancer cells (1–10 mM)[53]. Thus, we investigated the upper limit of the stability of the PC8 bioconjugate **11** in the presence of 1 mM GSH at pH 7.4. From the HPLC analysis, the bioconjugate **11** formed with the *N*-alkylpyridinium reagent reveal high stability at pH 7 for up to 1 day at room temperature (90% intact), even in the presence of elevated amounts of glutathione (1 mM). Over longer time course, 82% of PC8 conjugate **11** remained after 2 days incubation and decreased to 25% over 14 days, showing the displacement of compound **1** from the conjugate **11** via 1,6-elimination and thiol exchange with GSH (Supplementary Fig. 60). This could be an attractive mechanism for traceless release of molecular cargoes, in the cytosol of cancer cells that are known to have high GSH concentrations and valuable for biomedical applications.

### Site-selective modification of proteins

During protein modification, reagents are frequently used in large excess (10-fold or more) to enhance the conversion. However, this could compromise the specificity for thiol *versus* amino groups, as has been observed in the case of maleimides[31,32]. Therefore, the thiol selectivity of *N*-alkylpyridinium and maleimide reagents was compared when used in increasing equivalents with Tet peptide. The Tet peptide was selected as it contains a cysteine, a lysine residue and an *N*-terminal amine within its sequence. Notably, when up to 10 equiv of each bioconjugation reagent (compound **1** and maleimide shown in Supplementary Table 3) were applied, only the *N*-alkylpyridinium derivative **1** afforded the single-modified Tet conjugate. In contrast, the Michael addition reaction of maleimide and Tet resulted in a mixture of single and double-modified Tet.

The limits of excess reagent usage were then tested on a therapeutically relevant protein, the anti-HER2 (Trastuzumab) recombinant human monoclonal antibody (4D5-8), which is a full-length antibody. Specifically, the trifunctional *N*-alkylpyridinium **6** was used to functionalize anti-HER2 and demonstrate its potential application in the preparation of antibody-drug conjugates, as well as for comparison with classic maleimides. Anti-HER2 antibody was fluorescently labelled with sulfoCy5 dye via maleimide-sulfoCy5 and in a two-step reaction with *N*-alkylpyridinium **6** and TCO-sulfoCy5. Even when using a great excess of reagent **6** (20 equiv per Cys), a dye-to-antibody ratio of 0.6 was achieved based on UV/Vis quantification (Fig. 6a and Supplementary Fig. 92). The expected mass shifts of ~1493 Da over the two-step reaction in both the antibody's light chain (1470 Da) and heavy chain (1571 Da) were observed in the mass analysis by matrix-assisted laser desorption/ionization-time of flight (MALDI-ToF, Fig. 6b). A mass shift of 1490 Da was observed in the antibody's light chain in Liquid Chromatography-Electrospray Ionization-High Resolution MS (LC-ESI-HRMS) (Supplementary Fig. 93). The MS analyses indicate a single modification in the antibody over two reaction steps, which further confirm the chemoselectivity of the *N*-alkylpyridinium reagent. In contrast, when 2.5 equiv of maleimide per Cys was used, under similar reaction conditions reported by Jeon et al.[54], a dye-to-antibody ratio of around 8 was obtained based on MALDI-ToF and LC-ESI-HRMS (Supplementary Fig. 95–97). Critically, single and dual maleimide additions were observed in the mass spectra of the light chain of the antibody (764 Da per maleimide-sulfo-Cy5), due to an undesired cross-reaction, possibly with the α-amino group at the *N*-terminus. Moreover, using a larger excess of the same maleimide (20 equiv per Cys) led to a complete loss of thiol selectivity, resulting in a dye-to-antibody ratio around 25 and aggregation of the obtained bioconjugates (Supplementary Fig. 98). Therefore, the analogous maleimide used here is less chemoselective for thiol, requiring stricter stoichiometry of the reagent. Altogether, the results on the functionalization of peptides and full antibody via 1,6-addition to *N*-alkylpyridinium highlight the

higher thiol selectivity of *N*-alkylpyridinium over maleimide reagents, even when a large excess of reagent is required for selective protein bioconjugation.Nevertheless, due to the inherent interchain covalent (by disulfide bridges) and strong non-covalent interactions present in native antibodies[55,56], highly reactive reagents like maleimides remain as a better option when a high degree of modification at multiple sites is intended.

Next, the functionalization of other biologically relevant proteins containing a single and accessible cysteine residue with *N*-alkylpyridinium reagents was accomplished (Fig. 6c). Ubiquitin, a small regulatory protein that plays an important role in protein degradation and that contains a cysteine residue at its K63 position was selected for site-selective modification using the bioconjugation reagents **1**, **5** or **6**. Ubiquitin K63C (1 equiv) was incubated with compounds **1**, **5** (8 equiv) or **6** (35 equiv) in PB (50 mM, pH 7) overnight to ensure efficient modification. ESI-HRMS analysis showed successful modification of ubiquitin and different functionalities have been introduced successfully (Fig. 6d, SI section 9.1). To validate the site selectivity towards the cysteine residue on the ubiquitin surface, the solvent accessible thiol group (K63C) was masked with 4-DPS, which prevented any further modification, even when compound **1** was used in 30 equiv excess (Supplementary Fig. 83–84).

To further demonstrate the general applicability of our functionalization method, the anti-MMR nanobody targeting the macrophage mannose receptor in tumor-associated macrophages[57–60], was selected for modification with compound **6** under similar conditions as previously used for ubiquitin. The successful modification was confirmed by the correlation of the observed and calculated molecular mass (found: 15617 Da, calculated: 15619 Da) as shown in Fig. 6e. In addition, the C3bot1 toxin (**C3**) from *Clostridium botulinum* was also selected for dual modification using compound **6** to incorporate the bioorthogonal pairs for late-stage functionalization. ESI-HRMS data in Fig. 6f confirmed the successful site-selective modification to afford **C3-N₃-Tz** conjugate (found 25035 Da, calculated: 25038 Da). Noteworthy, CD spectra of the various modified proteins ubiquitin (K63C), anti-MMR nanobody and C3 enzyme indicate that the bioconjugation reaction did not disturb the native secondary structure of these proteins (Supplementary Figs. 85, 91 and 106).

Having demonstrated the successful introduction of two bioorthogonal groups into different protein substrates, the possibility of late-stage re-engineering of a native protein to confer additional properties beyond its natural function and mode of action was investigated. The C3 enzyme of *Clostridium botulinum* selectively ADP-ribosylates the small GTP-binding proteins Rho A, -B, and -C and inhibits their downstream signalling pathways[38,39,41], which blocks cell migration and has been applied for cancer applications[61–65]. However, the application of C3 is limited as it is poorly internalized into the cytosol of epithelial cells as compared to monocytic cells including monocytes, macrophages, osteoclasts and dendritic cells[40,41,66,67]. This greatly restricts the usage of native C3 protein for pharmacological inhibition of Rho-signalling in most cancer cells. [68] Thus, as a proof-of-concept, we used the *N*-alkylpyridinium reagent **6** for late-stage dual functionalization of Cys-containing C3 to enhance its delivery into A549 lung cancer cells and to track its uptake and fate within these cells. In this way, orthogonal reactions can be achieved in a sequential one-pot fashion to incorporate two payloads. Specifically, a combination of two different payloads was attached to the purified **C3-N₃-Tz** via two orthogonal click reactions as shown in Fig. 6g. The cell-penetrating RGD peptide was selected to redirect and mediate the uptake of C3 into cancer cells expressing integrin $\alpha_v\beta_3$ receptor, while the Cy5 served as a fluorophore for simultaneous tracking of the uptake of the resultant conjugate through confocal microscopy. The peptide (RGDC-DBCO) and the fluorophore (Cy5-TCO) were both attached to *N*-alkylpyridinium at Cys-13 of C3 enzyme, in order to achieve dual functionalization via two bioorthogonal reactions, i.e., the SPAAC and

iEDDA reaction respectively, affording **C3-RGD-Cy5** (found: 27110 Da, calculated: 27112 Da) in a one-pot reaction. The molecular mass increased (2075 Da) over the three reaction steps is in agreement with the calculated values (Fig. 6h). As a negative control, a FRET pair (Cy3 and Cy5 dyes) was also introduced to C3 sequentially via two orthogonal reactions (Cy3-DBCO via SPAAC and Cy5-TCO via iEDDA reaction) to afford the dual-functionalized protein conjugate **C3-Cy3-Cy5** (full chemical structure in SI) with the molecular weight of 26885 Da (Fig. 6i). The signal observed at 24775 Da over the 3 steps corresponds to the unmodified C3, which occurs due to fragmentation in the conditions of ESI-MS measurement. The same fragmentation pattern was also observed for the purified peptide conjugates (see for example Supplementary Fig. 25 concerning the ESI-MS analysis of PC8 conjugate **11** modified with compound **1**).

Thereafter, we applied both dual-modified C3 proteins to A549 lung cells, which overexpress the integrin $\alpha_v\beta_3$ receptor[69] and investigated the uptake of the C3 proteins into the cells via confocal microscopy. As shown in Fig. 7a, the functionalization of C3 with RGD now promoted efficient cell internalization of **C3-RGD-Cy5** in comparison to the control **C3-Cy3-Cy5**. Based on this observation, the effect of the different chemical modifications of C3 in vitro was investigated in more detail. It is well-established that C3 internalization into cellular cytosol can lead to intoxication, showing highly characteristic morphological cells changes with shrunken/rounded cell bodies and the formation of protrusions[70]. Thus, the cell morphology was investigated after incubation of A549 cells with various C3 substrates using phase contrast microscopy (Fig. 7b). After 5 h of treatment with two concentrations of the C3 substrates, **C3-RGD-Cy5** showed the most efficient uptake and intoxication (Fig. 7b–c) compared to the negative controls (**C3** and **C3-Cy3-Cy5**). The difference was most pronounced at 300 nM indicating a concentration-dependent effect (Fig. 7c). The efficient internalization of **C3-RGD-Cy5** into the cytosol of the cells was further confirmed by the biochemical analysis of the ADP-ribosylation status of Rho from these cells (Fig. 7d), using Western blot analysis of cell lysates after incubation with C3 substrates (see SI for detailed description of analysis). A strong band indicates that Rho was not or only slightly modified in the intact cells during toxin treatment, while a weak band indicates ADP-ribosylation by C3. As shown in Fig. 7d, strong bands were observed in the Western blot of untreated (Ctrl), native **C3** and **C3-Cy3-Cy5** treated cells. In contrast, the cells intoxicated by **C3-RGD-Cy5** showed a weaker signal for Biotin-ADP-ribosylated Rho in the same Western blot. **C3-RGD-Cy5** was more efficiently internalized into the cytosol of the cells and the dual-functionalized C3 retained its catalytic function to exhibit ADP-ribosyltransferase activity in the cytosol, which is in agreement with earlier results[64,68,70]. The reaction with the trifunctional *N*-alkylpyridinium reagent preserved the catalytic activity of C3 and significantly enhanced cytosolic uptake into cancer cells, resulting in intracellular ADP-ribosylation, thus successfully transferring the catalytic properties of C3 into cancer cells.

## Discussion

In this study, *N*-alkylpyridinium reagents were introduced as fast, selective and versatile tool for the chemoselective dual functionalization of peptides and proteins. Based on these features, these bioconjugation reagents are particularly suitable for diverse applications in chemical biology and biomedicine. The trifunctional *N*-alkylpyridinium reagent with two reactive bioorthogonal handles was obtained in a convenient synthesis in good yields based on commercially available starting materials. *N*-alkylpyridinium reagents are very stable and undergo a 1,6-addition of thiols providing excellent chemoselectivity, regioselectivity and high modification efficiency even in the presence of other nucleophiles. They offer a timely alternative to conventional maleimide reagents when dual modification at a single site with high chemoselectivity is intended, as summarized in Fig. 8.

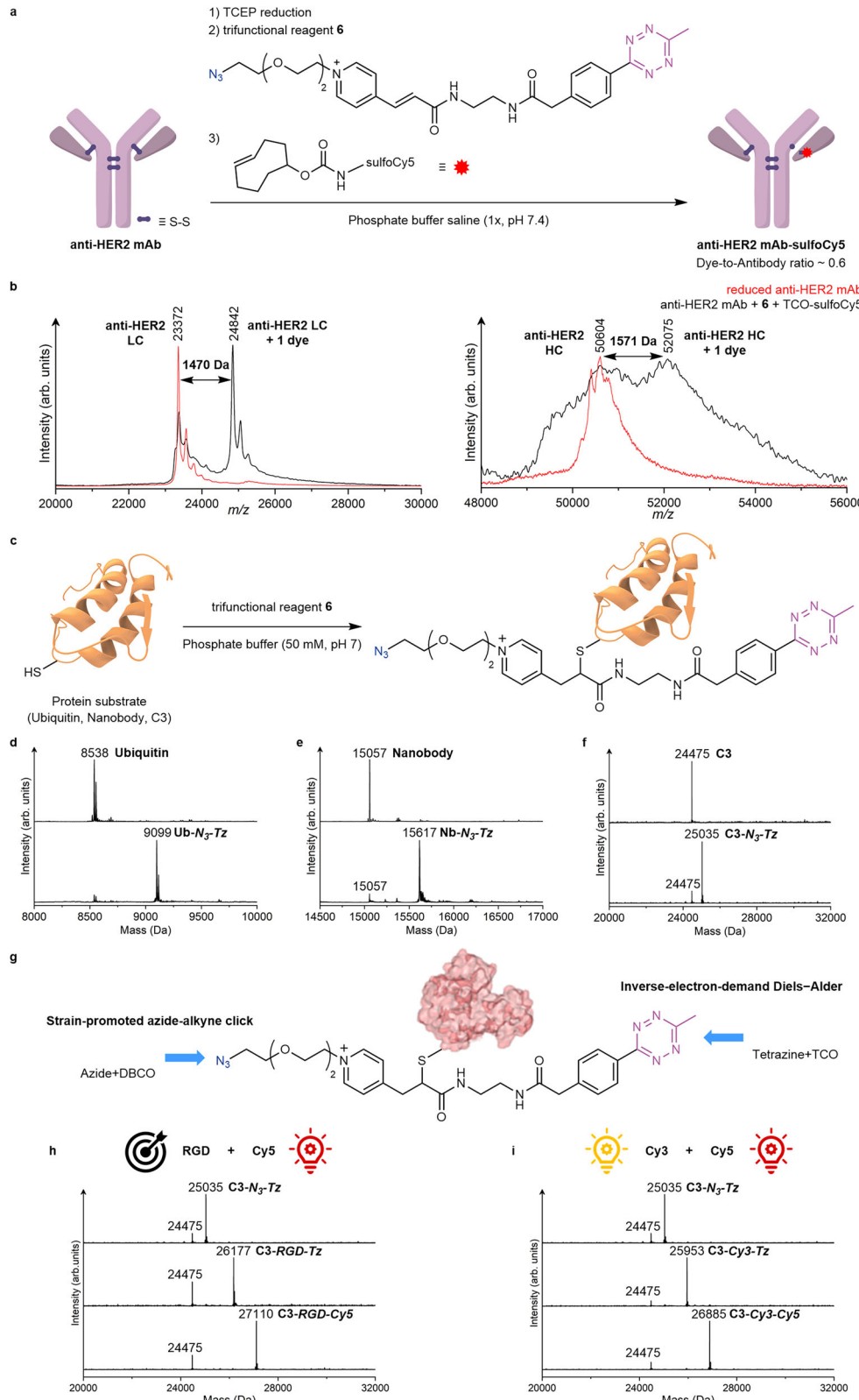

Bioconjugation with *N*-alkylpyridinium derivatives proceeds under mild, aqueous reaction conditions without side products. Compared to the less stable maleimides, excellent chemoselectivity is achieved even if the reagents are used in large excess, which is beneficial for protein conjugation (Fig. 8). Furthermore, the bioconjugates are compatible with European Medicines Agency (EMA) recommended storage in solution (for up to 24 h at low temperature)[71,72]. As a proof of

concept, we demonstrated the functionalisation of bacterial C3 enzyme with trifunctional *N*-alkylpyridinium carrying two bioorthogonal handles in a simple one-pot reaction under physiological conditions. The dual-modified C3 enzyme was transformed beyond its native function, now enabling its uptake into the cytosol of cancer cells, where it remodeled the cytoskeleton, as well as fluorescence labelling for cell studies.

**Fig. 6 | Scope of protein substrates. a** anti-HER2 (Trastuzumab) recombinant human monoclonal antibody (anti-HER2 mAb) fluorescently labelled via two-step reaction with compound **6** and TCO-SulfoCy5 dye. Reaction conditions: anti-HER2 mAb (1 equiv), Tris(2-carboxyethyl)phosphine (TCEP) for disulfide bridges reduction (5 equiv, 37 °C, 1 h), compound **6** (160 equiv, overnight, 20 °C) and TCO-sulfoCy5 (10 equiv, 1 h, 20 °C). **b** MALDI-ToF of reduced **anti-HER2 mAb** and **anti-HER2 mAb-sulfoCy5**, light chain (LC) and heavy chain (HC). Calculated mass shift: 1493 Da per N$_3$-Pyr-sulfoCy5 unit, found for LC: 1470 Da; HC: 1571 Da. Native anti-HER2 mAb LC, calculated 23442 Da, found 23372 Da; native anti-HER2 mAb HC calculated 50619 Da, found 50604 Da; anti-HER2 LC-N$_3$-Pyr-sulfoCy5 calculated 24935 Da, found 24842 Da; anti-HER2 HC-N$_3$-Pyr-sulfoCy5 calculated 52112 Da, found 52075 Da. **c** General scheme of site-selective modification of proteins with compound **6**. **d** Deconvoluted ESI-MS of **Ub-N$_3$-Tz** (calculated: 9101 Da, found: 9099 Da). Reaction conditions: Ub (1 equiv) and compound **6** (30 equiv) were mixed and incubated for 4 h at r.t.. **e** Deconvoluted ESI-MS of **Nb-N$_3$-Tz** (calculated:

15619 Da, found: 15617 Da). Reaction conditions: Nb (1 equiv) and compound **6** (40 equiv) were mixed and incubated overnight at r.t. **f** Deconvoluted ESI-MS of **C3-N$_3$-Tz** (calculated: 25038 Da, found: 25035 Da). Reaction conditions: C3 (1 equiv) and compound **6** (20 equiv) were mixed and incubated for 5 h at 20 °C. **g** Dual functionalization of **C3-N$_3$-Tz** conjugate with pairs of different functionalities (RGD, Cy3 and Cy5). C3 toxin schematic adapted from ref. 75, under a CC BY 4.0 license, https://creativecommons.org/licenses/by/4.0/. **h** Deconvoluted ESI-MS of **C3-RGD-Tz** (calculated: 26180 Da, found: 26178 Da), and **C3-RGD-Cy5** (calculated: 27112 Da, found: 27110 Da). Reaction conditions: **C3-N$_3$-Tz** (1 equiv) and RGDC-Dibenzocyclooctyne (DBCO) (10 equiv) were mixed and incubated for 4 h at 20 °C. Then, Cy5-trans-cyclooctene (TCO) (10 equiv) was also added and incubated for 1 h. **i** Deconvoluted ESI-MS of **C3-Cy3-Tz** (calculated: 25956 Da, found: 25953 Da), and **C3-Cy3-Cy5** (calculated: 26888 Da, found: 26885 Da). Reaction conditions: **C3-N$_3$-Tz** (1 equiv) and Cy3-DBCO (10 equiv) were mixed and incubated for 4 h at 20 °C. Then, Cy5-TCO (10 equiv) was also added and incubated for 1 h.

In contrast to conventional bioconjugation strategies or genetic encoding of noncanonical amino acids[73,74], we have demonstrated that dual functionalized bioconjugates can be easily obtained in situ and "in one-pot" with high efficiency, making it a user-friendly option also for non-chemists. Therefore, *N*-alkylpyridinium reagents will be useful to the broad scientific community to expand the arsenal of customized "precision" biotherapeutics to better target the rapidly evolving diseases of the future.

## Methods

### General procedure for the synthesis of *N*-alkylpyridinium derivatives

Commercially available compound (*E*)-3-(pyridin-4-yl)acrylic acid (1 equiv) was dissolved in DMF (10 mg/mL). After that, EDC·HCl (1.2 equiv) and DMAP (0.12 equiv) were added and the resultant mixture was stirred at rt for 30 min. Then, the amine-containing substrate (1.2 equiv) was added and the reaction mixture was stirred overnight at rt. After that, the solvent was evaporated under reduced pressure and the crude product was purified by flash column chromatography to get compound that contains the first functionality. Thereafter, the alcohol-containing substrate (1 equiv) was dissolved in anhydrous DCM, followed by the addition of 2,6-dimethyl pyridine (1.5 equiv). After that, trifluoromethane sulfonic anhydride (1.5 equiv) was added at 0 °C and the resultant mixture was stirred for 30 min. Without further purification, the resultant triflyl derivative was added to the aforementioned compound and the mixture was stirred overnight at rt. Thereafter, the reaction mixture was purified by HPLC to get the respective *N*-alkylpyridinium derivatives which were characterized by ¹H-NMR, ¹³C-NMR, ESI-LRMS and HRMS.

### Reaction profile of *N*-alkylpyridinium with nucleophiles

Compound **1** was chosen as a model compound to react with a thiol-containing substrate **7** and an amine-containing substrate **8** to investigate the reactivity of *N*-alkylpyridinium derivatives towards thiol and amino groups. Compound **1** (50 µg, 0.14 µmol, 1 equiv) was also incubated with compound **7** (35.6 µg, 0.30 µmol, 2.1 equiv) and **8** (41.4 µg, 0.30 µmol, 1.5 equiv) in 50 µL ACN:PB (50 mM, pH 7) (*v/v* = 2:3) mixture. The resultant mixture was incubated at rt for 4 h. Then the mixture was injected to the HPLC to monitor the reaction. In addition, compound **1** (50 µg, 0.14 µmol, 1 equiv) was also incubated with different amino acids, such as tyrosine (71.6 µg, 0.40 µmol, 2.8 equiv), methionine (59 µg, 0.40 µmol, 2.8 equiv), arginine (69 µg, 0.40 µmol, 2.8 equiv), histidine (61.3 µg, 0.40 µmol, 2.8 equiv), tryptophan (80.7 µg, 0.40 µmol, 2.8 equiv) and aspartic acid (53 µg, 0.40 µmol, 2.8 equiv), separately. All the reaction mixtures were incubated at rt for 4 h. Thereafter, 10 µL of mixture was injected to HPLC to check if compound **1** reacted with these amino acids or not.

### General procedure for site-selective modification of peptides

To a solution of peptide (1–2 mM) in 50 mM PB, pH 7 was added 1.2 equiv of compound **1** and mixed at rt. After 4–5 h, the conjugates were isolated by semi-preparative HPLC from the crude reaction mixtures with using a Zorbax Eclipse XDB-C18 HPLC column (80 Å, 9.4 × 250 mm, 5 µm) at a flow rate of 4 mL/min with acetonitrile (solvent A, containing 0.1% *v/v* TFA) and Milli Q water (solvent B, containing 0.1% *v/v* TFA). The modified peptide conjugates were characterized by ESI-LRMS and ESI-HRMS in positive mode.

### General procedure for site-selective modification of proteins

A solution of the protein of interest (1 mg/mL, 1 equiv) was prepared in 50 mM PB buffer, pH 7. Next, the corresponding *N*-alkylpyridinium reagent (8–40 equiv depending on reagents and the protein itself) was added and the resultant mixture was incubated overnight at rt. After that, the reaction mixture was purified by using amicon ultrafiltration tube to remove the excess bioconjugation reagents and organic solvent with using water as exchange solvent. The obtained protein conjugate was characterized by ESI-HRMS.

### Procedure for functionalization of trastuzumab

anti-HER2(4D5-8) (6.4 mg/mL in PBS 1x, pH 7.4) (6.0 µL, 0.27 nmol) was diluted to 2 mg/mL in 19.20 µL PBS 1x, pH 7.4 and TCEP (0.29 mg/mL, in 1 mM) (1.4 µL, 1.4 nmol) was added and shaken for 1 h at 37 °C. Then, compound **6** (25 mg/mL in MQ-water) (1.1 µL, 43 nmol) was added and the resultant mixture was incubated for 16 h at 20 °C. Then, the reaction mixture was transferred to ultrafiltration tube (0.5 mL, 10 kDa, PES filter) to remove the excess compound **6**, TCEP and organic solvent using PBS 1x pH 7.4 as exchange solvent (6x, 12000 G, 10 min). Thereafter, the purified conjugate solution was transferred to a new eppendorf and Sulfo-Cy5 TCO·TEA (25 mg/mL, 4.7 mM in water/1% DMSO) (0.52 µL, 2.7 nmol) was added and mixed for 1.5 h at 37 °C. Then, the mixture was purified by ultrafiltration using PBS pH 7.4 as exchange solvent (10x, 12000 G, 10 min) prior to MALDI-ToF analysis.

### Procedure for dual functionalization of C3 enzyme

To the Cys-C3bot1 solution (0.372 mg/mL in 50 mM PB, pH 7.4) (300 µL, 0.0045 µmol), compound **6** (25.0 mg/mL in DMF) (20 equiv, 2.00 µL, 0.090 µmol) was added and the resultant mixture was incubated for 5 h at 20 °C. The reaction mixture was transferred to a ultrafiltration tube (0.5 mL, 10 kDa, PES filter) to remove the excess compound **6** and organic solvent using 50 mM PB, pH 7.4 as exchange solvent (3x, 12000 G, 10 min) to deliver **C3-N$_3$-Tz** conjugate. The obtained **C3-N$_3$-Tz** was characterized by ESI-HRMS to confirm the successful modification.

A solution of RGDC-DBCO (20 mM in 50 mM PB, pH 7.4) (10 equiv, 0.75 µL, 0.015 µmol) was added, to the previously purified **C3-N$_3$-Tz**

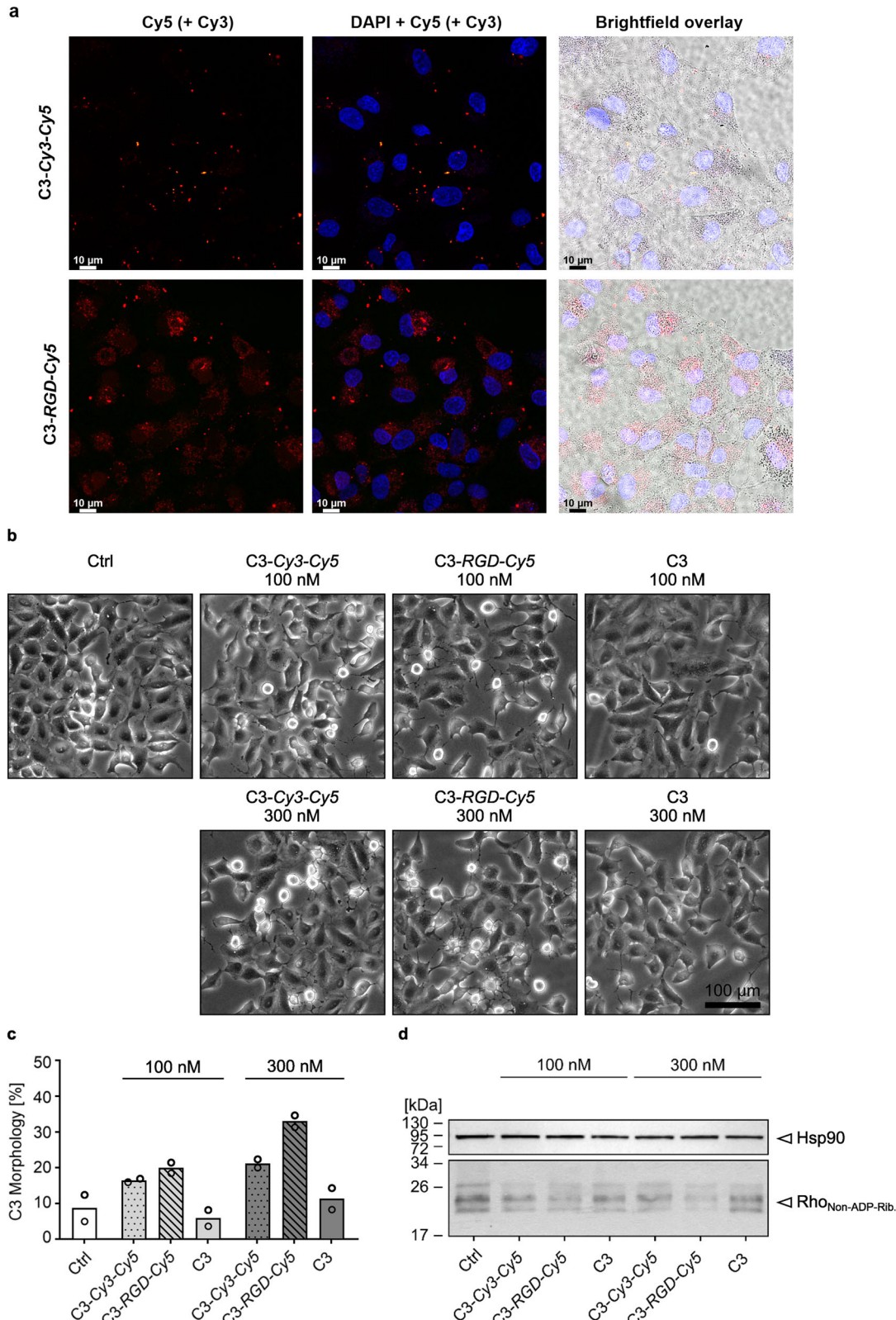

conjugate solution (100 μL) and mixed for 4 h under same conditions. Then, Cy5-TCO·TEA (5 mg/mL in H₂O/1% DMSO) (10 equiv, 3.18 μL, 0.015 μmol) was added and mixed for 1 h. After confirming the successful C3 dual modification (*m/z* 27110) by ESI-HRMS, the reaction mixture was purified by UF in a 6 mL tube (6x, 12000 G, 5 min) using 50 mM PB pH 7.4 as exchange solvent to afford the **C3-*RGD*-Cy5** conjugate.

### General statement for cell experiments

All experiments to analyze the effect of C3 toxin on cells were performed with commercially available cell lines (A549 cells). Therefore, as a standard sample size in this field of research, these experiments were performed four independent times with duplicates of each condition per experiment. This sample size was chosen according to previous studies in the same area of research[41,66,67].

**Fig. 7 | Cellular uptake and intoxication by dual-modified C3 proteins.**
**a** Fluorescence confocal microscopy showing uptake of C3 protein conjugates, **C3-Cy3-Cy5** and **C3-RGD-Cy5** into A549 cells treated with 300 nM of the respective C3 variants for 24 h and co-stained with NucBlue. Each experiment was repeated independently two times with similar results. DAPI channel at 405 nm, Cy5 channel at 649 nm and Cy3 channel at 554 nm. Scale bar: 10 μm. **b**–**d** A549 cells were treated with 100 nM or 300 nM of **C3, C3-Cy3-Cy5** or **C3-RGD-Cy5** at 37 °C for 5 h or left untreated (Ctrl). **b** Phase contrast microscopic images of intoxicated A549 cells. Scale bar: 100 μm. **c** Quantification of cells showing C3-morphology as percentage

of total cell number. Values are given as mean of duplicates within one experiment. **d** Intracellular Rho ADP-ribosylation status of cells after treatment for 8 h. Non-modified Rho in each cell lysate was biotin-labelled in the sequential ADP-ribosylation reaction by C3. Biotin-ADP-ribosylated Rho was detected in a Western blot by streptavidin-peroxidase conjugate, where a weak signal correlates with a high ADP-ribosyltransferase activity of C3 conjugates in living cells. Hsp90 served as loading control (n = 4). Detailed experimental information is provided in SI. **c, d** Source data are provided as a Source Data file.

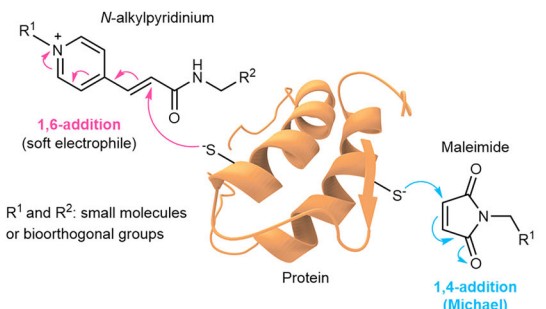

| Property | *N*-alkylpyridinium | Maleimide |
|---|---|---|
| Chemoselectivity | +++ | ++ |
| Reactivity | ++ | +++ |
| Reagent stability | +++ | + |
| Product stability | ≈ | ≈ |
| Aqueous solubility | +++ | + |
| Easy synthesis | = | = |
| Dual modification | ++ | + |

\* ≈ similar   = equal   + low   ++ medium   +++ high

**Fig. 8 | Summary comparing *N*-alkylpyridinium *vs* maleimide chemistry in protein bioconjugation.** + is marked in bold where property is more favorable.

## Confocal microscopy

A549 cells were seeded at a density of $2 \times 10^5$ cells per well in DMEM (10% FBS, 1% penicillin/streptomycin) within an IBIDI8-well confocal slide. After adhering for 24 h, cells were treated with 300 nM of **C3-RGD-Cy5** and **C3-Cy3-Cy5** for 24 h at 37 °C, 5% $CO_2$. After incubation, cells were washed thrice with PBS, co-stained with NucBlue and fixed with 4% PFA. Cells were imaged with a Leica Stellaris 8 microscope HC PL APO CS2 40x/1.25GLYC objective, 405 nm at 1.0% laser power for NucBlue excitation, 554 nm at 0.6% for Cy3 and 649 nm at 2.6% for Cy5. The experiment was repeated once with similar results.

## Intoxication studies

A549 cells were seeded in a 24-well microtiter plate with a cell number of $5 \times 10^5$ cells per ml 48 h prior treatment and incubated at 37 °C, 5% $CO_2$. They were then treated with 100 nM or 300 nM of Cys-C3bot1 (**C3**), **C3-Cy3-Cy5**, **C3-RGD-Cy5**, respectively or left untreated as control. Phase contrast microscopy was used to monitor changes in cell morphology after intoxication.

## Reporting summary

Further information on research design is available in the Nature Portfolio Reporting Summary linked to this article.

## Data availability

Supplementary information, cartesian coordinates (xyz), energies and relevant properties of all DFT optimized structures (Supplementary Data 1) and source data are provided in this paper. All datasets are available via Zenodo at https://doi.org/10.5281/zenodo.15209549. Source data are provided with this paper.

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

## Acknowledgements

T. W., H. B. and M.J.S.A.S. thank the DFG (CRC 1279 (C01)—project number 316249678) and L. X. thanks the National Science Foundation of China (project number: 22304181) for financial support. M.J.S.A.S. thanks the Max Planck Society for a postdoctoral scholarship. J.A.S.C. thanks the Fundação para a Ciência e a Tecnologia (FCT) for project PTDC/QUI-QOR/1786/2021 and Scientific Employment Stimulus 2020/02383/CEECIND. N.S. and J.B. are members of IGradU. We thank Prof. Lutz Nuhn and Prof. Jo A. Van Ginderachter for providing the anti-MMR nanobody, Prof. Zhixuan Zhou for discussion of the NMR interpretation, the Biocore facility for the confocal imaging and Dr. Fernando Bergamini for discussion on DFT calculations.

## Author contributions

L.X.: Initiating the project and conceptualization of the chemical design. Conducting synthetic and experimental work, performing 1,6-addition of thiols to the peptide and protein substrates, data analysis and processing, preparing figures and drafting the manuscript. M.J.S.A.S: Performing dual post-functionalization of C3 toxin and characterization of dual modified C3 conjugate, repeating 1,6-addi-tions of thiols to peptides and proteins to finalize analysis, performing and analyzing comparative stability studies, planning and designing confocal uptake studies for C3 toxin. Data analysis and processing, drafting the manuscript. J.A.S.C.: DFT calculations and interpretation, drafting discussion of DFT results. J. B. and N. S.: Conducting experimental work, data analysis and processing for biochemical/biological evaluation of activity of C3 toxin, drafting the discussion for biological evaluation of activity of C3 toxin. H. B.: Involved in the design and analysis of the biochemical/biological evaluation for C3 toxin, correcting the discussion regarding the biological aspects, supervision of J.B. and S.L.K.: Discussion of the design, concept, and results, correcting the manuscript, initiated collaboration for the project. T. W.: Involved in the inception and design concept, dis-cussion of the concept and results, correcting the manuscript, acquired funding for the project, supervision of L.X. and M.J.S.A.S.

## Funding

## Competing interests

The authors declare no competing interests.
