## [Transparent Peer Review file · Nature Communications]

Chemoselective Dual Functionalization of Proteins *via* 1,6-Addition of Thiols to Trifunctional *N*-Alkylpyridinium

Corresponding Author: Professor Tanja Weil

Version 0:

Reviewer comments:

Reviewer #1

(Remarks to the Author)

The manuscript by Xu and co-workers describes the 1,6-thiol addition of trifunctional *N*-alkylpyridinium as a novel site-selective bioconjugation strategy for the modification of cysteine residues. Using the Fukui index, determined by DFT, the authors were able to rationally design an *N*-alkylpyridinium reagent which would undergo regioselective 1,6-addition. The chemoselectivity for thiolates was determined in peptide studies, indicating a high degree of selectivity over primary amines (lysine and *N*-terminal amine). Having demonstrated the chemoselectivity of the bioconjugation approach, the authors applied it to cysteine containing proteins and confirmed efficient modification with negligible structural perturbation of the protein occurring. Utilising the trifunctional nature of the reagent described, the approach was subsequently applied in the generation of a C3 derivative which could penetrate A549 lung cancer cells upon attachment of the RGD cell-penetrating peptide improving its uptake in cells which don't uptake native C3, broadening the therapeutic potential of C3. Overall, this is a nice study which identifies a novel bioconjugation strategy and subsequently demonstrates its utility in the exploring the activity of modified C3 *in vitro*. I believe it should be published with minor corrections outlined below.

1. Page 1 line 30 "raised" should be change to "rose" and "in 2030" to "by 2030"
2. Figure 1 – please define C3 and RGD in the figure or the caption
3. Figure 3c. shows the maleimide reaction modelled but this is not discussed in the main text. Please include some discussion on this.
4. Page 9 - line 18. The isolated yields are stated which is good. Was the conversion quantitative?
5. Page 9 - line 29 Authors state 18% degradation after 45 h of compound 11 in 1 mM GSH. How does this compare to the equivalent maleimide under the same conditions? Is it an improvement? It would be beneficial if the authors could carry out this experiment. In the context of an ADC based on IgG, have much longer half-lives of days to weeks. It would be good if the authors could demonstrated stability of their conjugates and maleimides over a longer time course.
6. Page 10 – line 10 In the figure associated with table S1, double modification is indicated as lysine modification. Is the peptide *N*-terminally acetylated, can the authors rule out modification at the *N*-terminus? The lower pKa of *N*-terminal amines can make the reaction more favourable.
7. Figure 5 - Quality of the structures in this figure seem poor. The text is also slightly blurry. This is the case in all figures containing chemical structures. I would advise checking the resolution.
8. Figure 5f and 5g show LC-MS traces for modified species. However, all contain a peak at around 24500 Da. Is this unmodified C3. Please label this and comment on why the reaction is not quantitative.

Reviewer #2

(Remarks to the Author)

The authors present an approach for site-specific bioconjugation of cysteine residues in peptides and proteins using *N*-alkylpyridinium reagents. These reagents undergo efficient 1,6-addition with cysteine thiols, facilitated by the electron-deficient nature of the pyridinium moiety. The synthetic accessibility and versatility of *N*-alkylpyridinium derivatives enable the preparation of trifunctional reagents, expanding their utility in bioconjugation applications. A comprehensive computational study was provided to explain the high selectivity and efficiency.

Michael addition reaction has been widely used for Cys-specific bioconjugation on proteins, with maleimide being the most popular choice. There are also recent reports focusing on using different strategies to explore further potential with Michael addition (J. Am. Chem. Soc. 2024, 146, 3, 1776–1782; J. Am. Chem. Soc. 2017, 139, 17, 6146–6151). The current work reported falls in the same category. Thus, the novelty of the current work is moderate.

The author compared their new method with maleimide in the paper and showed maleimide could have addition product on sites beyond Cys, shown in table S1. However, when they used 2 equivalents of maleimide, the conversion was 100%. The reactivity of maleimide is significantly higher than their reported reagent 1, so maleimide really doesn't need a long time and high equivalents for the quantitative and selective Cys conjugation. This experiment cannot demonstrate reagent 1 is superior to maleimide.

The drawback of maleimide is that it is an unstable product that can go through retro-michael addition to decompose (~50%). The conjugation product reported here in this paper has a bit better stability, but there is still 15%-30% decomposition; this is not a minor amount.

Functionalization of different proteins using this method was demonstrated in this work. In general, the conjugation reactions show specificity; however, the conversion yield overall is not high compared with many other Cys-based conjugation reactions since many Cys-based conjugations give quantitative conversions.

1. Fig. S9.5, ubiquitin reactivity with 5 is relatively low, only 70% conversion.
2. ~10% ubiquitin is still left unreacted in Fig. 5b.
3. Fig. S9.12, Fig. 5c, ~15% left over; Mass seems noisy, what are the other peaks?
4. Fig. S9.16, Fig. 5d, ~30% left over; not working very well here.
5. Fig. S9.19, ~45% conversion, even lower conversion. Fig. S9.19 raw data and Fig. 5f deconvoluted data don't seem to match.
6. Fig. S9.20, Cy5 was attached after Cy3; if Cy3 step conversion is only 45%, why, after two-step reactions conversion went up higher to 80% ???
7. Fig. S9.22, ~55% conversion, still quite low. Fig. S9.22 raw data and Fig. 5g deconvoluted data don't seem to match.

Minor comments:

Specific reaction conditions should be listed in the figure legend, such as Fig. 5.

Page 11. Line 12, Fig. 5b should be Fig. 5c. Line 14, Fig. 5c should be Fig. 5d.

Based on the comments listed above, I don't think this manuscript is suitable for publication at Nature Communications.

Reviewer #3

(Remarks to the Author)

Reviewer #4

(Remarks to the Author)

The authors describe the preparation and application of trifunctional N-alkylpyridinium reagents for site-selective modification of proteins with two different payloads. The design of the most 1,6-selective N-alkylpyridinium reagents is guided by calculation of the Fukui indices of the substrates, while DFT calculations of the Gibbs energy profile of the reaction are used to gain a deeper understanding of this transformation.

The manuscript is clear and well written, and the calculations are a nice support to the experimental results presented. However, there are a couple of points that could be improved, regarding the calculations:

- Although the free energy profiles of Figure S7.3 clearly show that thiol 1,6-addition is kinetically favored over thiol 1,4-addition (by 6.5 kcal/mol), information about the thermodynamics of the reaction is lacking, as the real products of both thiol 1,6- and 1,4-additions, the corresponding N-alkylpyridinium ions or salts, are not shown in these reaction profiles. In Fig S7.3B, the final product is an enolate, which will always be less stable than the protonated final product. The same is applicable to FigS7.3c.

Which is thermodynamically more stable, the N-alkylpyridinium compound resulting from 1,6-addition or the one resulting from 1,4-addition? I feel this information is relevant to the present study, because the reactions are probably under thermodynamic control, and this should be included in the paper.

- In Figure S7.2, both LUMO+1 and the HOMO are not relevant to the discussion. Instead, it would be more interesting to see the representation of the relevant HOMO (HOMO-1 or HOMO-2).

Minor comments

- I would change “1,6-thiol addition” to “thiol 1,6-addition” or “1,6-addition of thiols” throughout the text, as I feel it is clearer. Also, it might be helpful to indicate the positions 1 and 6 (and 1 and 4) in the corresponding molecules (for example in Scheme 1b), so that the nomenclature is clear to everyone, especially for non-chemists.
- The prefix n in n-octanol (page 4, line 25) is not recommended by either IUPAC or CAS/SciFinder. The accepted names are octan-1-ol (IUPAC) or 1-octanol (CAS/SciFinder).
- Page 5, text below Scheme 1: change “quaternization of the Nitrogen” to “quaternization of the nitrogen atom”.
- Page 5, text below Scheme 1: change “1,4- addition reaction respectively” to “1,4-addition reaction, respectively”.
- Page 5, line 8: change “ACN:PB” to “acetonitrile:phosphate buffer (ACN:PB)” as there is no acronym index.
- Page 6, line 19: change “pH” to “pH values”.
- Change “methylthiolate” to “methanethiolate” throughout the manuscript.

Version 1:

Reviewer comments:

Reviewer #1

(Remarks to the Author)

The authors have improved the manuscript significantly by clarifying the questions raised adding new text, providing additional references and/or adding additional data. The thorough study and efficiency of 1,6-thiol addition of trifunctional N-alkylpyridinium method for site-selective cysteine bioconjugation warrants publication of this manuscript in Nature Communications in its current form.

Reviewer #2

(Remarks to the Author)

I appreciate the authors' efforts in addressing my and other reviewers' comments.

When I evaluate bioconjugation papers, I look at two main aspects. If the reaction novelty is high, then I don't believe it necessarily needs to achieve exceptional efficiency or product stability. However, if the reaction novelty is moderate, it should outperform current standards for papers published here. In the case of Cys-based conjugations, there are numerous published papers that have already demonstrated high efficiency, excellent product stability, and other achievements. For new Cys-based reaction, if the mechanism is entirely novel, then fewer additional goals need to be met. But if the reaction mechanism is already widely used, I pay closer attention to the outcomes.

For these reasons, I still consider the current work to be a minor advancement compared to maleimide conjugation. In the end, both methods involve Michael addition on Cys residues. While the product of the current conjugation is more stable than that of maleimide, as demonstrated by the authors, the stability difference is marginal and not significant.

Regarding selectivity, it's essential to consider reactivity as well. Maleimide is highly reactive, which means it requires less material and shorter reaction times. The N-alkylpyridinium derivative, being less reactive, can tolerate more material over longer reaction times, but does that actually lead to better relative selectivity? Maleimide addition has been applied to many commercial ADC preparations, with numerous studies showing that it can react with the eight exposed Cys residues on an antibody. When all eight Cys are exposed and excess maleimide is used, a near-homogeneous product with a DAR of ~7.8 can be achieved (as seen with T-DXD), indicating quite selective reactivity on antibodies. I am uncertain how this N-alkylpyridinium derivative would perform on antibodies and whether it would be significantly better than maleimide.

For product analysis, if fragmentation is due to ionization (which is common), the authors could run a longer LC-MS gradient to rule out the presence of unreacted starting material. If fragmentation occurs prior to MS analysis, the conjugate and native protein should show some level of separation on HPLC, particularly for small proteins. If fragmentation indeed happens in MS, there should always be a single conjugate elution peak on HPLC. Therefore, displaying all HPLC peaks and their corresponding MS peaks could definitively clarify product purity and reaction efficiency.

Again, I appreciate the authors' efforts in addressing these comments, but my opinion remains the same.

Reviewer #4

(Remarks to the Author)

The authors have satisfactorily addressed all my questions and concerns. I believe the manuscript is now suitable for publication in its current form.

Version 2:

Reviewer comments:

Reviewer #2

(Remarks to the Author)

I appreciate the authors' comparative analysis of trastuzumab reactivity. To facilitate the evaluation of expected mass data, the full sequence of trastuzumab should be provided in the Supporting Information (SI). It is standard practice to present expected and observed mass values side by side in the figure legend for clarity. Additionally, I am uncertain whether the authors have access to ESI-based mass spectrometry instruments, such as ESI-TOF or QTOF, as these could significantly enhance data quality. Ideally, the light chain should appear as a single peak, while the heavy chain should display three distinct peaks rather than a broad hump. If EndoS is used to trim glycans on the heavy chain, a single well-defined mass peak should be observed. I suggest improving this data quality is possible.

Regarding the reaction with reagent 6, both light and heavy chains exhibit incomplete modification. Specifically, ~30% of the light chain and ~50% of the heavy chain remain unreacted, even when a large excess of reagent 6 is used. The reason for this incomplete conversion is unclear to me.

In Figure RL5, the reaction of reagent 6 with ubiquitin shows a conversion of approximately 80% based on mass spectrometry data. Given the extensive literature on cysteine-specific modifications, this conversion rate is not high compared to commonly reported values.

Version 3:

Reviewer comments:

Reviewer #1

(Remarks to the Author)

I have checked the comments and responses. I am OK with the level of characterisation provided for the conjugates.

REVIEWER COMMENTS

We thank all the reviewers for their time and valuable suggestions on our manuscript. We have revised the main text and supporting information based on the comments and suggestions received and the point-by-point responses are shown below:

Reviewer #1 (Remarks to the Author):

The manuscript by Xu and co-workers describes the 1,6-thiol addition of trifunctional N-alkylpyridinium as a novel site-selective bioconjugation strategy for the modification of cysteine residues. Using the Fukui index, determined by DFT, the authors were able to rationally design an N-alkylpyridinium reagent which would undergo regioselective 1,6-addition. The chemoselectivity for thiolates was determined in peptide studies, indicating a high degree of selectivity over primary amines (lysine and N-terminal amine). Having demonstrated the chemoselectivity of the bioconjugation approach, the authors applied it to cysteine containing proteins and confirmed efficient modification with negligible structural perturbation of the protein occurring. Utilising the trifunctional nature of the reagent described, the approach was subsequently applied in the generation of a C3 derivative which could penetrate A549 lung cancer cells upon attachment of the RGD cell-penetrating peptide improving its uptake in cells which don't uptake native C3, broadening the therapeutic potential of C3. Overall, this is a nice study which identifies a novel bioconjugation strategy and subsequently demonstrates its utility in the exploring the activity of modified C3 in vitro. I believe it should be published with minor corrections outlined below.

We thank the reviewer for the positive feedback and valuable suggestions to further strengthen our manuscript. The point-by-point responses are listed below.

Question 1. Page 1 line 30 “raised” should be change to “rose” and “in 2030” to “by 2030”

Author response: We thank the reviewer for spotting this. As we have rewritten part of the introduction for better clarity, the paragraph containing this has been removed.

Question 2. Figure 1 – please define C3 and RGD in the figure or the caption

Author response: Thank you for the suggestion. The targeting peptide RGD and the C3 protein toxin have now been defined in the caption of Fig. 1, which is highlighted in yellow on page 3. “The tripeptide arginine-glycine-aspartic acid (RGD) has been selected as targeting motif that binds to the integrin receptors on the cancer cell’s surface. C3bot1 toxin (C3) from *Clostridium botulinum* that catalyzed ADP-ribosylation of Rho proteins was selected for dual-

modification with a fluorescent dye (Cy5) and a RGD peptide for *in vitro* imaging and concomitant inhibition of specific Rho-mediated cellular pathways.”

Question 3. Figure 3c shows the maleimide reaction modelled but this is not discussed in the main text. Please include some discussion on this.

Author response: We have now included the discussion about the calculation of maleimide conjugation on page 8, highlighted in yellow: “The calculated activation energy for maleimide conjugation with the same model thiol was determined to be 10.8 kcal/mol, which is comparable to the new modification strategy reported here.”

Question 4. Page 9 - line 18. The isolated yields are stated which is good. Was the conversion quantitative?

Author response: We have determined the high conversion of the representative peptide “WSC02” based on a calibration curve, and a conversion of 84% has been obtained, which is consistent with the reported isolated yield (86%). The detailed determination is shown below as well as in SI (Fig. S8.10–Fig. S8.15).

Fig. S8.14 Chromatograms of the reaction mixture after 4h, WSC02 peptide and bioconjugate with Fmoc-Phe-OH (internal standard).

Table S2 Calculations for conversion of WSC02 peptide to WSC02 bioconjugate after 4h reaction in 50 mM PB pH 7.0 at 25°C.

Reaction time	WSC02 area	IS area	AUC ratio	[WSC02] (mg/mL)	Conversion (%)
4h	228599	2831906	0.080723	0.156	84

For PC8 and RGDC, we optimized the HPLC method using a longer analytical column (Zorbax Eclipse XDB-C18 column, 4.6 × 250 mm, 5 μm) to monitor the progress after 4h, to ensure that both PC8 and RGDC are better retained on the column and not in the solvent front. We determined the conversion of PC8 of 76% based on a stronger signal in the UV-VIS (see figure below). However, as RGDC itself gives a rather weak and broad signal, a meaningful conversion cannot be determined (Figure S8.17). These results are included in the SI.

Fig. S8.4 HPLC analysis of the reaction mixture of PC8 with compound 1 after 4h at 25°C (A), isolated PC8 bioconjugate 11 (B) and fresh solution of PC8 peptide (C) with Fmoc-Phe-OH as internal standard (214 nm detection).

Table S1 Calculations for conversion of PC8 peptide to PC8 bioconjugate 11 after 4h reaction in 50 mM PB pH 7.0 at 25°C.

Reaction time	PC8 area	IS area	AUC ratio	[PC8] (mg/mL)	Conversion (%)
4h	909601	4429057	0.205371	0.237	76

Furthermore, we have also included the chromatograms obtained during the purification process of each bioconjugate in the SI (Fig. S8.2, Fig. S8.7, Fig. S8.12, Fig. S8.17, Fig. S8.22, Fig. S8.27), which only show the peaks of the respective bioconjugate as single product and the excess of compound 1, indicating that no side products were formed.

The discussion regarding the high conversions of the peptides can be found on pages 9-10, highlighted in yellow: "Other peptides such as the integrin receptor binding peptide RGDC, CEIE peptide, chemokine receptor 4 antagonist WSC02 (sequence in Fig. 4), antimicrobial peptide Tet (sequence in Fig. 4) and pan-coronavirus (CoV) fusion inhibitor EK1C (sequence in Fig. 4) with different lengths, sequences and cysteine positions were also successfully

modified with **1** under mild reaction conditions and isolated in moderate to high yields (57–95%, Fig. 4d, SI section 8.1). We further determined the conversion of the peptides (76%–quantitative), which show only a single product formed and no side products. The conversions observed (PC8: 76%; WSC02: 84%) are in good agreement with the isolated yields (PC8: 65%; WSC02: 86%). Our results clearly demonstrate the general applicability of *N*-alkylpyridinium reagents for site-selective modification of peptides at the cysteine residue.”

Question 5. Page 9 - line 29 Authors state 18% degradation after 45 h of compound **11** in 1 mM GSH. How does this compare to the equivalent maleimide under the same conditions? Is it an improvement? It would be beneficial if the authors could carry out this experiment. In the context of an ADC based on IgG, have much longer half-lives of days to weeks. It would be good if the authors could demonstrate stability of their conjugates and maleimides over a longer time course.

Author response: We thank the reviewer for the important question on the stability of the bioconjugates. We will first briefly comment on the (1) stability of the *N*-alkylpyridinium bioconjugation reagent versus maleimide, then discuss (2) the stability of the bioconjugates and (3) highlight differences in the degradation pathways of both classes of bioconjugates as there are significant differences.

(1) Maleimides are known to undergo ring-opening reactions that can already occur before thiolation (J. Am. Chem. Soc. 1955, 77, 14, 3922–3923; J. Pharmaceutical Sciences, 1984, 73, 12, 1767-1771) and the resultant maleic amides are unreactive to thiols. This is a limitation if exact quantities of the maleimide have to be used for bioconjugation under stoichiometric control or if extended reaction time is needed when targeting less accessible thiols i.e. on proteins. The trifunctional *N*-alkylpyridinium bioconjugation reagents are stable and in a 24 hours experiment, we did not observe any degradation products (Fig. S5.1–5.3). A short discussion has been added into the manuscript.

(2) For direct comparison of the stability of the *N*-alkylpyridinium and maleimide bioconjugates, we have prepared PC8 bioconjugate (**11**) and an analogous PC8 maleimide bioconjugate and tested their stability at different pH and in the presence of 1 mM GSH (SI Section 8.2). We are aware that 1 mM GSH is rather high and does not reflect the low GSH concentrations in plasma (1-6 μ M) that antibody-drug conjugates are exposed to. However, many peptides and proteins function in cells, where GSH concentrations are considerably higher.

Both bioconjugates, **11** and PC8-maleimide, reveal high stability (> 80%) at pH 7 for 3 days (Figures R1 and R2). Over extended incubation times or under hydrolytic conditions, i.e. at pH 8, the PC8-maleimide bioconjugate undergoes the known ring opening reaction forming a stable product, which is shown in Figure R1. This data is

in good agreement with the reported literature, where ~70% of the thiosuccinimide conjugates are hydrolyzed via ring opening, and present thiol exchange (Figure S8.40) or retro-Michael to a lower extent (*Bioconj. Chem.* **22**, 1946–1953 (2011); *Angew. Chem. Int. Ed.* **52**, 12592-12596 (2013) and *Bioconj. Chem.*, **26**, 145–152 (2015)).

The stability of the PC8 bioconjugate (**11**) has also been tested for up to 14 days, pH 6–8, using HPLC analysis (Fig. R2) and the results are mostly comparable to maleimide bioconjugates. The new *N*-alkylpyridinium bioconjugate is most stable at pH 7 and after 24 h (72 h) at room temperature only < 10 % (< 20%) degradation was observed. Next, the stability of bioconjugate **11** was tested over 2 weeks under more physiological conditions relevant in cancer cells (1 mM GSH, DPBS, pH 7.4) using HPLC analysis (Fig. S8.38, Fig. R2 below). **11** appeared to be relatively stable for up to 2 days with only ~20% displacement of PC8.

- (3) The major degradation mechanisms of maleimides and *N*-alkylpyridinium under the conditions discussed above are different. The PC8-maleimide conjugate gradually forms the stable, ring-opened product in solution, which is a new, stable bioconjugate as depicted below in Fig. R1. In contrast, bioconjugate **11** undergoes 1,6-elimination when incubated over extended time periods (> 2 days) under neutral to basic conditions forming back the native peptide, which could be an interesting feature for controlled release in the future. For the sake of clarity, we have now changed the discussion in the main text (page 10).

a Stability of PC8-maleimide conjugate at different pHs (pH = 6, 7 and 8)

b Stability of PC8-maleimide at pH 6

time	Percentage of PC8 conjugate remained
0 h	100%
6 h	97%
12 h	92%
1 day	93%
2 days	91%
3 days	92%
7 days	82%
14 days	73%

c Stability of PC8-maleimide at pH 7

time	Percentage of PC8 conjugate remained
0 h	100%
6 h	97%
12 h	95%
1 day	92%
2 days	84%
3 days	81%
7 days	55%
14 days	29%

d Stability of PC8-maleimide at pH 8

time	Percentage of PC8 conjugate remained
0 h	100%
6 h	102%
12 h	81%
1 day	63%
2 days	36%
3 days	25%
7 days	7%
14 days	0%

Fig. R1 Stability studies of PC8-maleimide bioconjugate at three different pHs (pH = 6, 7 and 8).

a Stability of PC8 conjugate (11) at different pHs (pH = 6, 7 and 8)

b Stability of 11 at pH 6

time	Percentage of PC8 conjugate remained
0 h	100%
7 h	90%
12 h	87%
1 day	86%
2 days	81%
3 days	80%
7 days	73%
14 days	65%

c Stability of 11 at pH 7

time	Percentage of PC8 conjugate remained
0 h	100%
6 h	100%
12 h	99%
1 day	93%
2 days	88%
3 days	84%
7 days	65%
14 days	47%

d Stability of 11 at pH 8

time	Percentage of PC8 conjugate remained
0 h	100%
6 h	83%
12 h	74%
1 day	67%
2 days	65%
3 days	59%
7 days	45%
14 days	10%

Fig. R2 Stability studies of PC8 bioconjugate at three different pHs (pH = 6, 7 and 8).

Stability of PC8 conjugate in 1 mM GSH

time	Percentage of PC8 conjugate remained
0 h	100%
6 h	95%
12 h	93%
1 day	86%
2 days	80%
3 days	73%
7 days	45%
14 days	24%

Fig. S8.38 Stability studies of PC8 bioconjugate in the presence of 1 mM GSH, in 1x DPBS pH 7.4.

Stability of PC8-maleimide conjugate in 1 mM GSH

Fig. S8.40 Stability studies of PC8-maleimide bioconjugate in presence of 1 mM GSH, in 1x DPBS pH 7.4.

Question 6. Page 10 – line 10 In the figure associated with table S1, double modification is indicated as lysine modification. Is the peptide N-terminally acetylated, can the authors rule out modification at the N-terminus? The lower pKa of N-terminal amines can make the reaction more favourable.

Author response: We thank the reviewer for the comment. The Tet peptide used in this work indeed contains an *N*-terminal amine and a lysine residue. The lower pKa of the *N*-terminal α -amino group (pKa \approx 6–8) compared to the lysine ϵ -amino group (pKa \approx 10) (*Anal. Biochem.* **18**, 248-255 (1967); *Commun. Chem.* **3**, 67 (2020); *R. Soc. Open Sci.* **9**, 211563 (2022) and *Chem. Soc. Rev.* **44**, 5495-5551 (2015)) makes the reaction with *N*-terminal amines more favorable. A similar result was also reported in the literature that the *N*-terminal was modified when a large excess of maleimide reagents were used for modification (*Nat Commun* **11**, 1015 (2020)). The figure in Table S3 (Table S1 in original submission) has already been revised accordingly which is also shown below.

Table S3 Comparison of the modification specificity of compound **1** and maleimide reagents when modifying Tet peptide, which contains a cysteine residue, a lysine residue and an N-terminal α -amino group in its backbone.

Substrate	Equiv.	Product distribution
		Cysteine/Double-modified
1	2 eq	100% / No detected
MI	2 eq	100% / No detected
1	10 eq	100% / No detected
MI	10 eq	67% / 33%

Question 7. Figure 5 - Quality of the structures in this figure seem poor. The text is also slightly blurry. This is the case in all figures containing chemical structures. I would advise checking the resolution.

Author response: We apologize for the low resolution that occurred during the PDF file conversion. We have provided all high-resolution figures separately to ensure they can be properly reviewed and considered.

Question 8. Figure 5f and 5g show LC-MS traces for modified species. However, all contain a peak at around 24500 Da. Is this unmodified C3. Please label this and comment on why the reaction is not quantitative.

Author response: The mass of the peaks in Fig. 5f and 5g corresponds to 24475 Da, which matches the mass of the unmodified C3. This is now labeled in Fig. 5f and 5g on page 14. The signal at 24475 Da that corresponds to the unmodified C3 is mainly due to fragmentation during the ESI-MS measurement. This is inferred from our observation that fragmentation occurred during the ESI-MS measurement for the isolated peptide bioconjugate modified with compound **1**. For example, for the isolated pure PC8 peptide bioconjugate, ESI-MS spectra showed both the signals of compound **1** at 253 Da and the unmodified PC8 peptide at 987 and 1009 Da (shown below, Fig. S8.3). Therefore, the peak intensity cannot be linearly

correlated to the amount of the proteins. We have included the information about possible fragmentation of the bioconjugate in the discussion now.

Fig. R3 ESI-MS analysis of PC8 bioconjugate modified with compound 1.

To further answer the reviewer's question and confirm the modification efficiency on the protein level, we calculated the degree of the labeling of the RGD- and Cy5-modified C3 Toxin (C3-RGD-Cy5) using the absorbance of the protein, DBCO and Cy5 dye. Around 71% of labeling efficiency can be achieved with C3 toxin over three sequential steps. The detailed calculation is included in the supporting information on page 82 as well as below.

Wavelength	Abs (a.u.)	Average	Calculated conc. (μM)
A280nm	0.335	0.345	corrected [C3 toxin]* 7.88
	0.363		
	0.338		
A309nm	0.141	0.131	
	0.129		
	0.123		
A650nm	1.514	1.401	5.6
	1.372		
	1.316		

*concentration correction
Org. Biomol. Chem., **2020**, *18*, 1140-1147
 SulfoCy5 dye CF(280nm) = 0.03
 ϵ_{280} (the extinction coefficient of Cy5 dye) = 250000 M⁻¹ cm⁻¹

Interchim - DQP580_DBCO CF
 DBCO CF(280nm) = 1.089
 ϵ_{309} (the extinction coefficient of DBCO) = 250000 M⁻¹ cm⁻¹

$$\text{Degree of labelling (DOL) of C3-RGD-Cy5} = \frac{A_{\text{max}} \times \epsilon_{280}(\text{protein})}{(A_{280} - A_{\text{max}} \times \text{CF}) \times \epsilon_{\text{max}}} = 71\%$$

Fig. S9.24 Calculation of the DOL of **C3-RGD-Cy5**.

Reviewer #2 (Remarks to the Author):

The authors present an approach for site-specific bioconjugation of cysteine residues in peptides and proteins using N-alkylpyridinium reagents. These reagents undergo efficient 1,6-addition with cysteine thiols, facilitated by the electron-deficient nature of the pyridinium moiety. The synthetic accessibility and versatility of N-alkylpyridinium derivatives enable the preparation of trifunctional reagents, expanding their utility in bioconjugation applications. A comprehensive computational study was provided to explain the high selectivity and efficiency.

Michael addition reaction has been widely used for Cys-specific bioconjugation on proteins, with maleimide being the most popular choice. There are also recent reports focusing on using different strategies to explore further potential with Michael addition (J. Am. Chem. Soc. 2024, 146, 3, 1776–1782; J. Am. Chem. Soc. 2017, 139, 17, 6146–6151). The current work reported falls in the same category. Thus, the novelty of the current work is moderate.

Author response: We appreciate the reviewer's critical feedback on our manuscript. Indeed, Michael reactions, particularly using maleimides, are extensively utilized for cysteine bioconjugation in protein chemistry.

We would like to clarify that the novelty of this work lies in the development of a new structurally simple and stable trifunctional N-alkylpyridinium bioconjugation reagent for chemoselective dual modification of proteins, which cannot easily be achieved with maleimides. Although maleimide reagents have been used for double payload modification at a single site, the preparation often involved multistep organic synthesis due to the limited reactive sites (R. Soc. Open Sci. 9, 211563 (2022) and see figures R4-R5 below). This is less appealing for non-chemists for broad applicability. On the other hand, the trifunctional N-alkylpyridinium reagent we report here can be synthesized over **two steps** with higher polarity and better water solubility (lower CLog P) compared to maleimide-containing trifunctional reagents (Chem. Sci. 11, 18-32-1838 (2020)) and dibromopyridazinedones that are reported (Nat. Commun. 6, 6645 (2015)). This is supported by the fact that all the dual modification reagents synthesized herein are soluble in aqueous solutions, even when more hydrophobic functionalities, such as strain-promoted alkyne or tetrazine groups are incorporated (for example, compounds 5 and 6 are completely soluble in Milli-Q water at 10 mM concentration and compound 1 at 50 mM concentration in 50 mM PB pH 7.0). The ease of synthesis and improved water solubility will allow broad applications in bioconjugation chemistry. For the sake of clarity, we have now strengthened the abstract and the introduction.

Editorial note: panel redacted

b. Synthetic scheme of the *N*-alkylpyridinium trifunctional reagent (this work)

Fig. R4 Synthetic scheme of maleimide-containing trifunctional reagent from the literature and the proposed *N*-alkylpyridinium trifunctional reagent developed in this work.

Editorial note: panel redacted

b. Synthetic scheme of *N*-alkylpyridinium trifunctional reagent (this work)

Fig. R5 Synthetic scheme of dibromopyridazinediones trifunctional reagent from the literature and the proposed *N*-alkylpyridinium trifunctional reagent developed in this work.

Moreover, in this study, we introduce a new 1,6-addition of thiols to *N*-alkylpyridinium derivatives as a distinct approach for dual modification of proteins in a site-selective manner.

This reaction is unprecedented and offers distinct advantages compared to the “conventional” 1,4-addition mechanism associated with maleimides and other typical Michael acceptors. The *N*-alkylpyridinium reagents bear a more extended conjugated system for 1,6-addition, which renders them as soft electrophiles. According to the hard-soft acid-base (HSAB) theory, *N*-

alkylpyridinium derivatives preferably react with soft nucleophiles, such as thiolates rather than with amines (hard nucleophiles) (shown below, Fig. R6). The experimental data indicates that our reagents offer better chemoselectivity compared to the conventional 1,4- thiol-maleimide conjugation (Table S3). In addition, we also performed new DFT calculations showing that the activation free energies (ΔG^\ddagger) for the reaction between *N*-alkylpyridinium 1 and thiolate ($\Delta G^\ddagger = 7.6$ kcal/mol) is much lower than that for amines ($\Delta G^\ddagger = 17.7$ kcal/mol) with a significant difference of 10.1 kcal/mol.

Fig. R6 (a) *N*-alkylpyridinium derivatives are considered as soft electrophiles, which prefer to react with soft nucleophiles, such as thiolates rather than amines (hard nucleophiles). (b) Gibbs free energy profile for the reaction between compound 1 and thiolate (c) the reaction between compound 1 and amines. DFT calculations were performed at the M06-2X/6-31+G(d,p)/PCM(water) level of theory (energy values in kcal mol⁻¹). Transition state geometries depict the N-C bond distance in Å.

Taken together, our work presents a new type of bioconjugation reagent for dual functionalization of proteins at cysteine residues, which (1) is easy to synthesize, (2) more stable than conventional maleimides which undergo hydrolysis, (3) offers fast and easy dual modification of proteins in water, (4) shows high chemoselectivity due to 1,6-addition addition compared to the classical 1,4-Michael addition, thus allowing (5) its usage in large excess without side products and is particularly attractive for protein modification (which is not possible with maleimides).

We have also restructured our introduction and discussion to highlight these important points and for better clarity of the novelty.

The author compared their new method with maleimide in the paper and showed maleimide could have addition product on sites beyond Cys, shown in table S1. However, when they used 2 equivalents of maleimide, the conversion was 100%. The reactivity of maleimide is significantly higher than their reported reagent 1, so maleimide really doesn't need a long time and high equivalents for the quantitative and selective Cys conjugation. This experiment cannot demonstrate reagent 1 is superior to maleimide.

Author response: We agree with reviewers that for the peptide substrates with lower molecular weight and simple 2D structure, bioconjugation typically achieves high efficiency with a slight excess of conjugation reagents. As demonstrated in Table S1 (now Table S3), both 2 equivalents of compound 1 and maleimide gave quantitative conversion when modifying the Tet peptide.

However, the study presented in Table S1 (now Table S3) is used to demonstrate conditions typically used for protein modification. Proteins often possess complex 3D structures where thiol groups are not fully accessible. Moreover, protein samples are typically prepared in rather diluted solutions (in the range of μM to nM) to prevent aggregation. Due to low reaction concentration, a higher excess of bioconjugation reagents, such as 10 equivalents (*Int. J. Pharm. X* **1**, 100020 (2019) and *Nat. Commun.* **7**, 13128 (2016)), is employed to ensure effective modification. Therefore, excellent chemoselectivity towards cysteine residues is mandatory to minimize undesired cross-reactions with *N*-terminus or lysine residues to yield homogeneous products that retain full bioactivity. In this case, we seek to demonstrate this point on a peptide model, which can be characterized more easily using MS, compared to a large protein. The results shown in Table S1 (now Table S3) thus show that the bioconjugate reagents reported in this work demonstrate superior thiol specificity over maleimide reagents when applied in excess conditions used in protein modification. We have now rephrased the discussion on page 11 in the main text for better clarity.

The drawback of maleimide is that it is an unstable product that can go through retro-Michael addition to decompose (~50%). The conjugation product reported here in this paper has a bit better stability, but there is still 15%-30% decomposition; this is not a minor amount.

Author response: We thank the reviewer for the important question on the stability of the bioconjugates. We will first briefly comment on the (1) stability of the *N*-alkylpyridinium bioconjugation reagent versus maleimide, then discuss (2) the stability of the bioconjugates and (3) highlight differences in the degradation pathways of both classes of bioconjugates as there are significant differences.

- (1) Maleimides are known to undergo ring-opening reactions that can already occur before thiolation (*J. Am. Chem. Soc.* 1955, **77**, 14, 3922–3923; *J. Pharm. Sci.*, 1984, **73**, 12, 1767-1771) and the resultant maleic amides are unreactive to thiols. This is a limitation if exact quantities of the maleimide have to be used for bioconjugation under stoichiometric control or if extended reaction time is needed when targeting less accessible thiols i.e. on proteins. The trifunctional *N*-alkylpyridinium bioconjugation reagents are stable and in a 24 hours experiment, we did not observe any degradation products (Fig. S5.1–5.3). A short discussion has been added into the manuscript.
- (2) For direct comparison of the stability of the *N*-alkylpyridinium and maleimide bioconjugates, we have prepared PC8 bioconjugate (**11**) and an analogous PC8 maleimide bioconjugate and tested their stability at different pH and in the presence of 1 mM GSH (SI Section 8.2). We are aware that 1 mM GSH is rather high and does not reflect the low GSH concentrations in plasma (1-6 μ M) that antibody-drug conjugates are exposed to (*Analyst* **140**, 3339-3342(2015)). However, many peptides and proteins function in cells, where GSH concentrations are considerably higher.
- Both bioconjugates, **11** and PC8-maleimide, reveal high stability (> 80%) at pH 7 for 3 days (Figures R1 and R2). Over extended incubation times or under hydrolytic conditions, i.e. at pH 8, the PC8-maleimide bioconjugate undergoes the known ring opening reaction forming a stable product, which is shown in Figure R1. This data is in good agreement with the reported literature, where ~70% of the thiosuccinimide conjugates are hydrolyzed via ring opening, and present thiol exchange (Figure S8.40) or retro-Michael to a lower extent (*Bioconj. Chem.* **22**, 1946–1953 (2011); *Angew. Chem. Int. Ed.* **52**, 12592-12596 (2013) and *Bioconj. Chem.*, **26**, 145–152 (2015)).
- The stability of the PC8 bioconjugate (**11**) has also been tested for up to 14 days, pH 6–8, using HPLC analysis (Fig. R2) and the results are mostly comparable to maleimide bioconjugates. The new *N*-alkylpyridinium bioconjugate is most stable at pH 7 and after 24 h (72 h) at room temperature only < 10 % (< 20%) degradation was observed. Next, the stability of bioconjugate **11** was tested over 2 weeks under more physiological conditions relevant in cancer cells (1 mM GSH, DPBS, pH 7.4) using HPLC analysis (Fig. S8.38, Fig. R2 below). **11** appeared to be relatively stable for up to 2 days with only ~20% displacement of PC8.
- (3) The major degradation mechanisms of maleimides and *N*-alkylpyridinium under the conditions discussed above are different. The PC8-maleimide bioconjugate gradually forms the stable, ring-opened product in solution, which is a new, stable bioconjugate as depicted below in Fig. R1. In contrast, bioconjugate **11** undergoes 1,6-elimination when incubated over extended time periods (> 2 days) under neutral to basic conditions forming back the native peptide, which could be an interesting feature for

controlled release in the future. For the sake of clarity, we have now changed the discussion in the main text (page 10).

a Stability of PC8-maleimide conjugate at different pHs (pH = 6, 7 and 8)

b Stability of PC8-maleimide at pH 6

time	Percentage of PC8 conjugate remained
0 h	100%
6 h	97%
12 h	92%
1 day	93%
2 days	91%
3 days	92%
7 days	82%
14 days	73%

c Stability of PC8-maleimide at pH 7

time	Percentage of PC8 conjugate remained
0 h	100%
6 h	97%
12 h	95%
1 day	92%
2 days	84%
3 days	81%
7 days	55%
14 days	29%

d Stability of PC8-maleimide at pH 8

time	Percentage of PC8 conjugate remained
0 h	100%
6 h	102%
12 h	81%
1 day	63%
2 days	36%
3 days	25%
7 days	7%
14 days	0%

Fig. R1 Stability studies of PC8-maleimide bioconjugate at three different pHs (pH = 6, 7 and 8).

a Stability of PC8 conjugate (11) at different pHs (pH = 6, 7 and 8)

b Stability of 11 at pH 6

time	Percentage of PC8 conjugate remained
0 h	100%
7 h	90%
12 h	87%
1 day	86%
2 days	81%
3 days	80%
7 days	73%
14 days	65%

c Stability of 11 at pH 7

time	Percentage of PC8 conjugate remained
0 h	100%
6 h	100%
12 h	99%
1 day	93%
2 days	88%
3 days	84%
7 days	65%
14 days	47%

d Stability of 11 at pH 8

time	Percentage of PC8 conjugate remained
0 h	100%
6 h	83%
12 h	74%
1 day	67%
2 days	65%
3 days	59%
7 days	45%
14 days	10%

Fig. R2 Stability studies of PC8 bioconjugate at three different pHs (pH = 6, 7 and 8).

Stability of PC8 conjugate in 1 mM GSH

time	Percentage of PC8 conjugate remained
0 h	100%
6 h	95%
12 h	93%
1 day	86%
2 days	80%
3 days	73%
7 days	45%
14 days	24%

Fig. S8.38 Stability studies of PC8 bioconjugate in the presence of 1 mM GSH, in 1x DPBS pH 7.4.

Stability of PC8-maleimide conjugate in 1 mM GSH

time	Percentage of PC8 conjugate remained
0 h	100%
6 h	99%
12 h	98%
1 day	93%
2 days	87%
3 days	83%
7 days	58%
14 days	34%

Fig. S8.40 Stability studies of PC8-maleimide bioconjugate in presence of 1 mM GSH, in 1x DPBS pH 7.4.

Functionalization of different proteins using this method was demonstrated in this work. In general, the conjugation reactions show specificity; however, the conversion yield overall is not high compared with many other Cys-based conjugation reactions since many Cys-based conjugations give quantitative conversions.

1. Fig. S9.5, ubiquitin reactivity with 5 is relatively low, only 70% conversion.
2. ~10% ubiquitin is still left unreacted in Fig. 5b.
3. Fig. S9.12, Fig. 5c, ~15% left over; Mass seems noisy, what are the other peaks?
4. Fig. S9.16, Fig. 5d, ~30% left over; not working very well here.
5. Fig. S9.19, ~45% conversion, even lower conversion. Fig. S9.19 raw data and Fig. 5f deconvoluted data don't seem to match.
6. Fig. S9.20, Cy5 was attached after Cy3; if Cy3 step conversion is only 45%, why, after two-step reactions conversion went up higher to 80% ???
7. Fig. S9.22, ~55% conversion, still quite low. Fig. S9.22 raw data and Fig. 5g deconvoluted data don't seem to match.

Author response to collective question from points 1-7: We thank the reviewer for the careful check and have made response according to questions with common points as follows.

Concerning to question 3: the others peaks with low intensity, which are not assigned in the combined ion series of anti-MMR nanobody come from incomplete reduction with TCEP.

Concerning data agreement in questions 5 and 7: To ensure that we have a better representation of the species in the MS, we took the full range as shown in the raw data, rather than focusing on narrow range for deconvolution. Therefore, this gives a better averaged, deconvoluted spectrum and the deconvolution was performed with MaxEnt1. The same procedure was applied to the range displayed in the combined ion series for all protein conjugates.

Concerning conversion rate in questions 1-7: Mass spectrometry (MS) is a powerful technique to enable the direct identification of molecules based on mass-to-charge ratios, as well as semi-quantitative analysis. However, we observed that the purified peptide bioconjugates undergo fragmentation (see figure below and Fig S8.3) in ESI-MS to generate ions of both peptide bioconjugate, as well as the native peptide and reagent 1.

Fig. R6 LC-MS analysis of the isolated PC8 conjugate **11** (calculated: 1239.3 [M]⁺, 620.7 [M+H]²⁺, 414.1 [M+2H]³⁺, 310.8 [M+3H]⁴⁺; 253.1 [Compound 1]⁺, 987.0 [PC8+H]⁺, 1008.7 [PC8+Na]⁺, found: 1238.9 [M]⁺, 620.2 [M+H]²⁺, 413.8 [M+2H]³⁺, 310.6 [M+3H]⁴⁺, 253.1 [Compound 1]⁺, 987.0 [PC8+H]⁺, 1008.7 [PC8+Na]⁺).

Furthermore, the hypothesis of fragmentation in the MS measurements was confirmed by additional MS analysis of azido- and tetrazine-modified C3 (C3-N₃-Tz) with different applied voltages. Even with a lower capillary voltage (2 vs 3 kV), as well as other ion source parameters like sampling cone (20 vs 40 V) and source offset (50 vs 80 V), we still observed fragmentation to different extents (see deconvoluted and raw MS spectra below). Therefore, the ion intensity cannot be directly correlated for the determination of the relative amount of native and modified peptide/protein conjugate. The driving force for this fragmentation comes from the formation of a product with a conjugated system, which is an inherent feature of these new reagents. Thus, the signal intensity for the different species (conjugates and precursors) do not reflect the reaction yields or relative amount of the mixture components, but rather the fragmentation probability during ESI measurement.

A Raw and deconvoluted ESI-MS data of C3-N₃-Tz
(Applied capillary voltage **3 kV**, sampling cone (CV) **40 V** and source offset (SO) **80 V**)

B Raw and deconvoluted ESI-MS data of C3-N₃-Tz
(Applied capillary voltage **2 kV**, sampling cone (CV) **40 V** and source offset (SO) **80 V**)

C Raw and deconvoluted ESI-MS data of C3-N₃-Tz
(Applied capillary voltage **2 kV**, sampling cone (CV) **20 V** and source offset (SO) **80 V**)

D Raw and deconvoluted ESI-MS data of C3-N₃-Tz
(Applied capillary voltage **3 kV**, sampling cone (CV) **20 V** and source offset (SO) **50 V**)

Fig. R7 Deconvoluted ESI-MS data of C3-N₃-Tz with MaxEnt1. Different capillary voltage (2 vs 3 kV), ion source parameters like sampling cone (20 vs 40 V) and source offset (50 vs 80 V) were applied for measurement.

Given the observed fragmentation in MS, the values from absorbance measurement are more reliable to determine the degree of modification. Thus, we calculated the degree of the labeling of the RGD- and Cy5-modified C3 Toxin (C3-RGD-Cy5) to confirm the modification efficiency. Around 71% of labeling efficiency can be achieved with C3 toxin over three sequential steps. Literature using trifunctional maleimide reagents show conjugation efficiency of 70–80% over 3 steps (*Bioorg. Med. Chem. Lett.* **28**, 3617-3621 (2018)). Therefore, a labeling efficiency of 71% over three sequential steps in our case is comparable to literature methods.

The detailed calculation is included in the supporting information on page 82 in SI as well as below.

Wavelength	Abs (a.u.)	Average	Calculated conc. (μM)
A280nm	0.335	0.345	corrected [C3 toxin]* 7.88
	0.363		
	0.338		
A309nm	0.141	0.131	
	0.129		
	0.123		
A650nm	1.514	1.401	5.6
	1.372		
	1.316		

*concentration correction
Org. Biomol. Chem., **2020**, *18*, 1140-1147
 SulfoCy5 dye CF(280nm) = 0.03
 ϵ_{280} (the extinction coefficient of Cy5 dye) = 250000 M⁻¹ cm⁻¹

Interchim - DQP580_DBCO CF
 DBCO CF(280nm) = 1.089
 ϵ_{280} (the extinction coefficient of DBCO) = 250000 M⁻¹ cm⁻¹

$$\text{Degree of labelling (DOL) of C3-RGD-Cy5} = \frac{A_{\text{max}} \times \epsilon_{280}(\text{protein})}{(A_{280} - A_{\text{max}} \times \text{CF}) \times \epsilon_{\text{max}}} = 71\%$$

Fig. S9.24 Calculation of the DOL of **C3-RGD-Cy5**.

Minor comments:

Specific reaction conditions should be listed in the figure legend, such as Fig. 5.

Author response: We thank the reviewer for the suggestions and the reaction conditions have been added to the figure caption of Fig. 5 which are highlighted in yellow on page 14.

Page 11. Line 12, Fig. 5b should be Fig. 5c. Line 14, Fig. 5c should be Fig. 5d.

Author response: We are sorry for the mistake. This is corrected and highlighted in the manuscript.

Based on the comments listed above, I don't think this manuscript is suitable for publication at Nature Communications.

Reviewer #3 (Remarks to the Author):

I co-reviewed this manuscript with one of the reviewers who provided the listed reports. This is part of the Nature Communications initiative to facilitate training in peer review and to provide appropriate recognition for Early Career Researchers who co-review manuscripts. We thank the reviewer for the time and appreciate the interest in reviewing our work.

Reviewer #4 (Remarks to the Author):

The authors describe the preparation and application of trifunctional N-alkylpyridinium reagents for site-selective modification of proteins with two different payloads. The design of the most 1,6-selective N-alkylpyridinium reagents is guided by calculation of the Fukui indices of the substrates, while DFT calculations of the Gibbs energy profile of the reaction are used to gain a deeper understanding of this transformation.

The manuscript is clear and well written, and the calculations are a nice support to the experimental results presented. However, there are a couple of points that could be improved, regarding the calculations:

We thank the reviewer for the positive feedback and valuable suggestions for this work. The detailed responses are shown below.

Question 1. Although the free energy profiles of Figure S7.3 clearly show that thiol 1,6-addition is kinetically favored over thiol 1,4-addition (by 6.5 kcal/mol), information about the thermodynamics of the reaction is lacking, as the real products of both thiol 1,6- and 1,4-additions, the corresponding N-alkylpyridinium ions or salts, are not shown in these reaction profiles. In Fig S7.3B, the final product is an enolate, which will always be less stable than the protonated final product. The same is applicable to FigS7.3C.

Which is thermodynamically more stable, the N-alkylpyridinium compound resulting from 1,6-addition or the one resulting from 1,4-addition? I feel this information is relevant to the present study, because the reactions are probably under thermodynamic control, and this should be included in the paper.

Author response: The DFT study was performed with methanethiolate as the nucleophile, resulting in the formation of the corresponding enolates as the direct products. This can be compared to the previously reported work by Bernardes group (*Nat. Commun.* **7**, 13128 (2016)), which studied other acceptors using the same DFT level of theory. Since enolates are not the final products of the experimental reaction, we performed the proton affinity (PA) study to obtain insights into the structure of the final protonated product (Fig. S7.4, page 24 in SI), which is also shown below. This study indicates that protonation at the carbon atom **6** is the most thermodynamically favored, leading to the re-formation of the N-alkylpyridinium ion. Therefore, the calculated PA value (251 kcal mol⁻¹) gives information on the thermodynamics of this step.

Fig. S7.4 DFT computed proton affinities (PA) for intermediate product obtained from 1,6-addition of methanethiolate anion to compound **1**. Optimized structures of the products resulting from protonation at different atoms are shown.

To further address the referee's comment, we also performed a computational study using methanethiol instead of methanethiolate as the nucleophile at the same DFT level of theory. The calculations show similar thermodynamics for the 1,6-addition and 1,4-addition (Fig. S7.5).

Fig. S7.5. Gibbs free energies and optimized structures of reagents and products for **A**) 1,4- and 1,6-addition of methanethiol to compound 1, **B**) 1,4-addition of methanethiol to maleimide. DFT calculations were performed at the M06-2X/6-31+G(d,p)/PCM(water) level of theory (energy values in kcal mol⁻¹).

Question 2. In Figure S7.2, both LUMO+1 and the HOMO are not relevant to the discussion. Instead, it would be more interesting to see the representation of the relevant HOMO (HOMO-1 or HOMO-2).

Author response: Thanks for the advice. The HOMO (HOMO-1 and HOMO-2) of the compound 1 are calculated and presented in Fig S7.2 in the supporting information as well as below.

Fig. S7.2 Representations of the HOMO-1, HOMO-2, HOMO, LUMO, LUMO+1 and LUMO+2 of compound 1. Isosurfaces were generated with a contour value of 0.08 a.u..

Minor comments

Question 3. I would change “1,6-thiol addition” to “thiol 1,6-addition” or “1,6-addition of thiols” throughout the text, as I feel it is clearer. Also, it might be helpful to indicate the positions 1

and 6 (and 1 and 4) in the corresponding molecules (for example in Scheme 1b), so that the nomenclature is clear to everyone, especially for non-chemists.

Author response: Thank you for the valuable suggestions. Now we have changed the “1,6-thiol addition” to “1,6-addition of thiols” highlighted in yellow in the manuscript to make it more clear for the readers. In addition, scheme 1b is also revised to show the number of the corresponding atoms to further clarify the respective 1,6- and 1,4-addition of thiols. The revised Scheme 1b is also shown here.

b Model reaction showing 1,6- and 1,4-addition of thiols, respectively

Scheme 1b Model reaction showing the 1,6- and 1,4-addition of thiols to compound 1, respectively.

Question 4. The prefix n in n-octanol (page 4, line 25) is not recommended by either IUPAC or CAS/SciFinder. The accepted names are octan-1-ol (IUPAC) or 1-octanol (CAS/SciFinder).

Author response: Thanks for the advice. “n-octanol” in the main text is changed to Octan-1-ol, which is highlighted in yellow on page 4.

Question 5. Page 5, text below Scheme 1: change “quaternization of the Nitrogen” to “quaternization of the nitrogen atom”.

Author response: “quaternization of the Nitrogen” has been changed to “quaternization of the nitrogen atom” accordingly, which is highlighted in yellow on page 4.

Question 6. Page 5, text below Scheme 1: change “1,4- addition reaction respectively” to “1,4-addition reaction, respectively”.

Author response: It has been revised accordingly on page 5 highlighted in yellow.

Question 7. Page 5, line 8: change “ACN:PB” to “acetonitrile:phosphate buffer (ACN:PB)” as there is no acronym index.

Author response: The full names of ACN and PB have been added instead of the abbreviation, which is highlighted in yellow on page 6.

Question 8. Page 6, line 19: change “pH” to “pH values”.

Author response: “pH” is changed to “pH values” on page 7.

Question 9. Change “methylthiolate” to “methanethiolate” throughout the manuscript.

Author response: Thank you for pointing this out. “methylthiolate” has been changed to “methanethiolate” throughout the whole manuscript and is also highlighted in yellow.

REVIEWER COMMENTS

We thank all the reviewers for their time and positive feedback on our manuscript. We have revised the main text and supporting information based on the comments and suggestions received from reviewer 2 and the point-by-point responses are shown below:

Reviewer #1 (Remarks to the Author):

The authors have improved the manuscript significantly by clarifying the questions raised adding new text, providing additional references and/or adding additional data. The thorough study and efficiency of 1,6-thiol addition of trifunctional N-alkylpyridinium method for site-selective cysteine bioconjugation warrants publication of this manuscript in Nature Communications in its current form.

We thank the reviewer for the positive feedback on our manuscript.

Reviewer #2 (Remarks to the Author):

I appreciate the authors' efforts in addressing my and other reviewers' comments. When I evaluate bioconjugation papers, I look at two main aspects. If the reaction novelty is high, then I don't believe it necessarily needs to achieve exceptional efficiency or product stability. However, if the reaction novelty is moderate, it should outperform current standards for papers published here. In the case of Cys-based conjugations, there are numerous published papers that have already demonstrated high efficiency, excellent product stability, and other achievements. For new Cys-based reaction, if the mechanism is entirely novel, then fewer additional goals need to be met. But if the reaction mechanism is already widely used, I pay closer attention to the outcomes.

Q1. For these reasons, I still consider the current work to be a minor advancement compared to maleimide conjugation. In the end, both methods involve Michael addition on Cys residues. While the product of the current conjugation is more stable than that of maleimide, as demonstrated by the authors, the stability difference is marginal and not significant.

A1: With all due respect, we strongly disagree with the reviewer's assessment that the 1,6-addition of thiols is not novel, and reviewers 1 and 3 also clearly indicated a high degree of chemical novelty. It is wrong to classify the addition of thiols to *N*-alkylpyridinium reagents as a classical Michael-type addition. The Michael-type or 1,4-addition follows a distinctively different reaction mechanism compared to the novel 1,6-addition of thiols, which provides improved chemoselectivity. It is the first example of a 1,6-addition of thiols for bioconjugation and the *N*-alkylpyridinium reagent outperforms the golden standard "maleimide" in this regard.

Even though both reactions involve an addition to a conjugated double bond, the *N*-alkylpyridinium reagent has greatly enhanced thiol chemoselectivity due to its soft electrophile character, whereas maleimides are harder electrophiles and therefore cross react with primary amines, like the α -amino group at the *N*-terminus, as discussed in our manuscript. The new bioconjugation strategy based on the 1,6-addition of thiols provides a simple and highly chemoselective reagent towards thiols that even allows dual modification in one reaction step.

Q2: Regarding selectivity, it's essential to consider reactivity as well. Maleimide is highly reactive, which means it requires less material and shorter reaction times. The *N*-alkylpyridinium derivative, being less reactive, can tolerate more material over longer reaction times, but does that actually lead to better relative selectivity? Maleimide addition has been applied to many commercial ADC preparations, with numerous studies showing that it can react with the eight exposed Cys residues on an antibody. When all eight Cys are exposed and excess maleimide is used, a near-homogeneous product with a drug-to-antibody ratio (DAR) of ~ 7.8 can be achieved (as seen with T-DXD), indicating quite selective reactivity on antibodies. I am uncertain how this *N*-alkylpyridinium derivative would perform on antibodies and whether it would be significantly better than maleimide.

A2: The balance of reactivity vs selectivity of bioconjugation reagents is crucial. We have determined that this balance is very different for the *N*-alkylpyridinium reagent compared to maleimides, which is a unique selling point of the 1,6-addition to thiols in our manuscript (the other one is the fast and simple dual modification). As discussed in our manuscript, the lower chemoselectivity of maleimides is due to their higher reactivity that can lead to side products and cross-reactions, i.e. with the α -amino group of the *N*-terminus of peptides, which has been observed for the Tet peptide (SI p. 67, Table S3) as well as the antibody's light chain of the clinically used protein trastuzumab.

To demonstrate the **higher chemoselectivity of *N*-alkylpyridinium compared to maleimide**, we have labeled the HER-2 (Trastuzumab) Recombinant Human Monoclonal Antibody (4D5-8) by applying the bioconjugation reagent **6** and fluorescently labeled with TCO-sulfoCy5 in a second reaction step. The *N*-alkylpyridinium reveals a significantly higher chemoselectivity in this head-to-head comparison compared to the corresponding reaction with maleimide-sulfoCy5 (see below in Fig. RL1 and Fig. 5e and 5f of main text). Most importantly, a much greater excess of *N*-alkylpyridinium of 20 equiv per cysteine has been applied to challenge the 1,6-addition reaction and remarkably, the obtained dye-to-antibody ratio remained "1" clearly indicating **single modification over two reaction steps**. Using the reagent in excess with chemoselectivity is one of the key features that allows chemical modification of biomolecules, which can even be extended to reactions on a solid support. Thus, the simple and inexpensive synthesis of *N*-alkylpyridinium further expands the chemoselective portfolio of bioconjugation strategies.

We have conducted a similar reaction based on the reaction conditions reported in the paper by Jeon *et al.* (*Bioorg. Chem.* 149, 107504 (2024)), i.e. the application of 2.5 equiv of maleimide per cysteine. Based on MALDI-ToF analysis, **a dye-to-antibody ratio of around 8 was obtained**, which has also been reported in the corresponding paper. In addition, also dual maleimide addition in the antibody's light chain (LC) was observed according to MALDI measurements (see below Fig. RL2 and Fig. S9.20), which is a result of the undesired cross-reaction with the α -amino group at the *N*-terminus. In case a large excess of this maleimide reagent is used (20 equiv per cys), we observe a complete loss of thiol selectivity leading to a dye-to-antibody ratio of around 25, which also leads to protein aggregation and loss of the bioconjugates indicated by low yields. In this control reaction, which has now been included in the SI (see below Fig. RL3 and Fig. S9.21), the maleimide-Cy5 failed to afford thiol selectivity, whilst *N*-Alkylpyridinium only showed a single modification in the LC. These new results are discussed in page 11 and 12 of the main text.

Although maleimides are a gold standard and are widely used across disciplines, there are still limitations that become critical when high chemoselectivity is required, such as in the case of therapeutic biomolecules, or when dual modification at a single site with high chemoselectivity is warranted. **We clearly show that the novel *N*-alkylpyridinium reagents are a great option to generate a pure bioconjugation product based on the chemoselective modification and functional diversity is introduced by the trifunctional *N*-alkylpyridinium, leading to a clean dual modification in a one-pot reaction.** Therefore, *N*-alkylpyridinium reagents are in line with the increasing demand for pure biotherapeutics in terms of translation and regulatory requirements. However, maleimides will remain the reagent of choice when a high degree of modification at multiple sites is required. We have provided a table to help guide the choice between these two reagents (see below Fig. RL4, Fig. 7 in the main text) and have included a critical discussion in the main text. We hope that this clarifies the reviewer's open questions.

Editorial note: panel redacted

Fig. RL1 “Fig 5. (e) anti-HER2 (Trastuzumab) recombinant human monoclonal antibody (anti-HER2 mAb) fluorescently labelled with sulfoCy5 via two-step reaction with compound 6 and TCO-SulfoCy5 dye. Reaction conditions: anti-HER2 mAb (1 equiv), TCEP for disulfide bridges reduction (5 equiv TCEP, 37°C, 1h), compound 6 (160 equiv, overnight, 20°C) and TCO-sulfoCy5 (10 equiv, 1h, 20°C). (f) MALDI-ToF of anti-HER2 mAb-sulfoCy5 (light chain (anti-HER2 LC) and heavy chain (anti-HER2 HC).”

Editorial note: panel redacted

Fig. RL2 “Fig. S9.20 MALDI-ToF analysis of modified trastuzumab (anti-HER2 mAb) with 20 equiv of Maleimide-sulfoCy5. Light chain (anti-HER2 LC), heavy chain (anti-HER2 HC).”

Editorial note: panel redacted

Fig. RL3 “Fig. S9.21 MALDI-ToF analysis of modified trastuzumab (anti-HER2 mAb) with 160 equiv of Maleimide-sulfoCy5. Light chain (anti-HER2 LC), heavy chain (anti-HER2 HC).”

Fig. RL4 “Fig. 7 Summary comparing *N*-alkylpyridinium vs maleimide chemistry in protein bioconjugation.”

Q3: For product analysis, if fragmentation is due to ionization (which is common), the authors could run a longer LC-MS gradient to rule out the presence of unreacted starting material. If fragmentation occurs prior to MS analysis, the conjugate and native protein should show some level of separation on HPLC, particularly for small proteins. If fragmentation indeed happens in MS, there should always be a single conjugate elution peak on HPLC. Therefore, displaying all HPLC peaks and their corresponding MS peaks could definitively clarify product purity and reaction efficiency.

A3: We repeated the MS analysis of the protein conjugates through LC-MS under denaturing conditions in addition to the direct infusion of desalted aliquots of the purified products, which has already been included in the previous revised version. We have included the UV/Vis and TIC chromatograms, along with the raw and deconvoluted MS in the SI (Fig. S9.4; S9.6; S9.8; S9.10; S9.16; S9.18; S9.24; S9.26). As shown in the example below (Fig. RL5), compound **6** has a different retention time from the obtained protein conjugate (6.76 min for remaining residual **6** vs 7.56 and 7.69 min for **Ub-N₃-Tz conjugate**). Therefore, although a fragmentation peak at *m/z* 561.2655 corresponding to reagent **6** was detected in the same retention time of Ub-N₃-Tz, this is due to the fragmentation that occurs during the ionization process.

Concerning to reaction efficiency, it is hard to quantify based on the chromatograms, as the native and modified protein have similar retention time. However, averaged raw (Fig. S9.10b) and deconvoluted MS (Fig. S9.10c) clearly indicate a high conversion and purity of the desired protein conjugates.

Fig. RL5. LC-MS analysis on the purified Ub-N₃-Tz conjugate: a) reaction scheme of Ub(K63C); 10 equiv of compound 6, 21h at 20 °C. b) UV/Vis (residual reagent 6, RT 6.8 min; Ub-N₃-Tz, RT 7.6 and 7.7 min); c) TIC; d) averaged MS of peak at RT 7.3–8.1 min (compound 6 calc. 561.6265, found: 561.2655) e) deconvoluted MS (Ub-N₃-Tz calc. 9101, found: 9100; Ub calc. 8540, found: 8541).

Again, I appreciate the authors' efforts in addressing these comments, but my opinion remains the same. We thank the reviewer time and critical thinking of our work. We hope all concerns have been addressed with the additional data included in the manuscript and SI.

Reviewer #4 (Remarks to the Author):

The authors have satisfactorily addressed all my questions and concerns. I believe the manuscript is now suitable for publication in its current form.

We thank the reviewer for the positive feedback on our manuscript.

REVIEWER COMMENTS

Reviewer #2 (Remarks to the Author):

I appreciate the authors' comparative analysis of trastuzumab reactivity.

Q1. To facilitate the evaluation of expected mass data, the full sequence of trastuzumab should be provided in the Supporting Information (SI).

A1: The sequence of HER-2 (Trastuzumab) Recombinant Human Monoclonal Antibody (clone 4D5-8) from Leinco Technologies (catalog number LT1500) used in this work is included in the SI, highlighted in yellow.

(Heavy chain)

EVQLVESGGGLVQPGGSLRLSCAASGFNKDTYIHWVRQAPGKGLEWVARIYPTNGYTRYADSVKGR
FTISADTSKNTAYLQMNSLRAEDTAVYYCSRWGGDGFYAMDYWGQGTLLTVSSASTKGPSVFPLAP
SSKSTSGGTAALGCLVKDYFPEPVTVSWNSGALTSQVHTFPAVLQSSGLYSLSSVVTVPSSSLGTQT
YICNVNHKPSNTKVDKVKVEPKSCDKTHTCPPCPAPELLGGPSVFLFPPKPKDTLMISRTPEVTCVVVD
VSHEDPEVKFNWYVDGVEVHNAKTKPREEQYNSTYRVVSVLTVLHQDWLNGKEYKCKVSNKALPAP
IEKTIKAKGQPREPQVYTLPPSREEMTKNQVSLTCLVKGFYPSDIAVEWESNGQPENNYKTTTPVLD
SDGSFFLYSKLTVDKSRWQQGNVFCFSVMHEALHNHYTQKSLSLSPG

(Light chain)

DIQMTQSPSSLSASVGDRVTITCRASQDVNTAVAWYQQKPKGKAPKLLIYSASFLYSGVPSRFSGRS
GTDFTLTISSLQPEDFATYYCQQHYTTPPTFGQGTKVEIKRTVAAPSVFIFPPSDEQLKSGTASVCLL
NNFYPREAKVQWVKVDNALQSGNSQESVTEQDSKSTYLSSTLTLSKADYEEKHKVYACEVTHQGLS
SPVTKSFRNGEC

(Disulfide bridge: H22-H96, H147-H203, H264-H324, H370-H428, H229-H'229, H232-H'232, L23-L88, L134-L194, H223-L214)

Q2. It is standard practice to present expected and observed mass values side by side in the figure legend for clarity. Additionally, I am uncertain whether the authors have access to ESI-based mass spectrometry instruments, such as ESI-TOF or QTOF, as these could significantly enhance data quality. Ideally, the light chain should appear as a single peak, while the heavy chain should display three distinct peaks rather than a broad hump. If EndoS is used to trim glycans on the heavy chain, a single well-defined mass peak should be observed. I suggest improving this data quality is possible.

A2: We would like to assure the reviewer that we have carefully checked the entire data set and all the expected and observed mass values are shown side by side in the legends of the figures. For MALDI-ToF-MS, the expected and observed mass shifts are included, which is more meaningful due to the resolution of the technique.

We would like to point out that we also analysed the antibody-dye conjugates using an ESI-QToF mass spectrometer (SYNAPT G2-Si, Waters Corp.), but neither the intact nor the modified heavy chain could be detected after addition of compound **6**. Thus, we used MALDI-ToF for characterization as the conjugate could be ionized and detected. Instead of EndoS, we used the PNGaseF Glycan Cleavage Kit (Gibco, catalogue number A39245 as given in the SI) for deglycosylation, which successfully deglycosylated the heavy chain of the native anti-HER-2 mAb and the conjugates, resulting in single peaks. The plots for the LC-ESI-HRMS analysis is provided in the SI (Figs. S9.21, S9.22, S9.24, S9.25, S9.27 and S9.28).

Q3. Regarding the reaction with reagent **6**, both light and heavy chains exhibit incomplete modification. Specifically, ~30% of the light chain and ~50% of the heavy chain remain unreacted, even when a large excess of reagent **6** is used. The reason for this incomplete conversion is unclear to me.

A3: For the anti-HER-2 mAb modified with compound **6** and TCO-sulfoCy5, a degree of labelling (DOL) of 0.6 was determined on the basis of Nanodrop UV/Vis measurements, which is in good agreement with the conversion as evidenced by MALDI-ToF and LC-ESI-MS data (see DOL calculation in SI Fig. S9.20).

We would like to emphasise that our *N*-alkylpyridinium is highly selective and reactive towards cysteines, as demonstrated for a broad range of peptides and proteins (CEIE, RGDC, PC8, WSC02, Tet and EK1C peptides in Fig. 4 and Ubiquitin, anti-MMR nanobody, C3 toxin proteins in Fig. 5c–i) and we achieved high degrees of functionalisation as demonstrated by the deconvoluted MS. In addition, we determined that about 71% of dual-functionalised C3-RGD-Cy5 was formed according to UV/Vis measurements, as provided in the previous and current version of the manuscript (Fig. 5c–i and Fig. S9.41).

We have a long-standing expertise in the area of peptide and protein modification and based on our experience, native antibodies represent a special case because they are stabilized by covalent (through disulfide bridges) and strong non-covalent interactions between their chains (*Biochemistry* **13**, 4602-4608 (1974) and *Immunochemistry* **14**, 45-52 (1977)). Highly reactive reagents such as maleimides allow a high degree of functionalisation of antibodies (3.4–8.0 fold) which, however, is not always desired (*Bioorg. Med. Chem. Lett.* **26**, 1542-1545 (2016), *Nat. Biotechnol.* **33**, 694-696 (2015)). Monofunctionalisation of native antibodies is not possible with maleimide reagents without careful control of the stoichiometry due to side reaction with *N*-terminal or lysine residues. Because of the different reaction mechanism, *N*-alkylpyridinium reagents are less reactive and therefore, they allow mono- or dual modification at a single cysteine site even when used in excess, which is not possible with maleimides. Moreover, maleimides are known to undergo side reactions with azides (*Org. Biomol. Chem.*, **7**, 3308-3318 (2009) and *J. Am. Chem. Soc.* **147**, 8049-8062 (2025)), rendering them incompatible for incorporation in a single, trifunctional reagent for dual functionalization of proteins. We are therefore convinced that *N*-alkylpyridinium reagents are an important alternative to maleimides, especially when precise mono- or dual functionalization at a single site is intended. Further discussion of these results is included in the manuscript (Figure 5 and 7, pages 11–12), also thanks to the constructive feedback raised by this reviewer.

Q4. In Figure RL5, the reaction of reagent 6 with ubiquitin shows a conversion of approximately 80% based on mass spectrometry data. Given the extensive literature on cysteine-specific modifications, this conversion rate is not high compared to commonly reported values.

A4: As discussed in the previous revision rounds, we observe a fragmentation of the obtained protein and peptide conjugates under ESI conditions (e.g. Fig. S8.3 for the PC8 conjugate and Fig. S9.10 for the Ubiquitin-N₃-Tz), which is not unusual and indicates that the relative peak intensity of the native protein vs. the modified protein is not reliable for quantification *via* this method. Nevertheless, in the deconvoluted mass spectra, we could clearly see the single modification of the various proteins (Ubiquitin, anti-MMR nanobody, C3 toxin proteins, Fig. 5c–i) and we could determine a high conversion rate (about 71%) for dual-functionalised C3-RGD-Cy5, Fig. S9.34) by UV/Vis quantification.

We would like to thank the reviewer for the constructive feedback and hope that we have been able to address the remaining concerns. All changes in the manuscript and SI are highlighted. Statements and data in main text and SI that are relevant for addressing the concerns are also highlighted to facilitate assessment.